∂ | Open Peer Review | Microbial Ecology | Research Article

# Inhibitory effects of berberine against *Streptococcus mutans*: an *in vitro* insight on its anticaries potential

Chongmai Zeng,[1] Wentao Jiang,[1,2] Chao Liu,[1] Rongcheng Yu,[1] Yang Li,[1] Peiru Li,[1] Yang Cao[1]

**ABSTRACT** Dental caries is a prevalent biofilm-related oral infectious disease, in which *Streptococcus mutans* plays an important role in the pathological process. The discovery of novel agents with inhibitory effects on *S. mutans* is essential to develop effective strategies against dental caries. Plant extracts are a rich source of novel chemicals for the discovery of more effective antimicrobial agents. *Coptis chinensis* Franch. is a traditional Chinese herbal medicine with various pharmacological properties. This study aimed to evaluate the antimicrobial activity of berberine (BBR), a bioactive alkaloid isolated from *Coptis chinensis* Franch., on the biofilm formation of *S. mutans* and to investigate the relevant mechanism of its action. The study found that BBR inhibited the growth of planktonic *S. mutans*, as demonstrated by Minimum Inhibitory Concentration, Minimum Bactericidal Concentration, and growth curve assay. BBR was found to inhibit biofilm formation and exhibit antimicrobial effects on *S. mutans*, as demonstrated by crystal violet staining assay, confocal laser scanning microscopy, and CFU counting assay. Moreover, the inhibitory effect of BBR on the virulence of *S. mutans* was evaluated by the anthrone-sulfuric method and lactic acid assay. Bacterial aggregation assay, membrane potential assay, and transmission electron microscopy were used to preliminarily elucidate the membrane-destructive bactericidal mechanism of BBR. The expression of virulence genes *gtfB*, *gtfC*, *gtfD*, *ldh*, *vicR*, *liaR*, and *comD* was measured using quantitative real-time PCR (qRT-PCR). The cell viability assay on human oral cells and macrophages demonstrated that BBR exhibited no cytotoxicity to the tested host cells. In conclusion, the study demonstrates that BBR has significant anticaries activity against *S. mutans* and has the potential to be further developed as a novel anticaries agent for clinical use.

**IMPORTANCE** Dental caries is a chronic disease affecting people worldwide, which could cause a series of oral problems. Plant-derived compounds have shown antimicrobial properties against a wide range of pathogens. This study investigated the antimicrobial effect of BBR on *Streptococcus mutans*, including the inhibitory effect on the planktonic bacteria growth, biofilm formation, virulence factors, and explored the mechanism and biocompatibility of BBR. This study demonstrated the significant anticaries activity of BBR against *S. mutans* and its potential to be further developed as a novel anticaries agent for clinical prevention and treatment of dental caries.

**KEYWORDS** *Streptococcus mutans*, berberine, biofilm, dental caries, natural products, anticaries

**Peer Reviewer** Marcia Dinis, UCLA - School of Dentistry, Los Angeles, California, USA

Address correspondence to Wentao Jiang, 834658081@qq.com, or Yang Cao, caoyang@mail.sysu.edu.cn.

Chongmai Zeng and Wentao Jiang contributed equally to this article. The author order was determined in order of increasing seniority.

The authors declare no conflicts of interest.

Dental caries is a chronic disease affecting people worldwide. Dental caries develops through a complex interaction over time between acid-producing bacteria, fermentable carbohydrates, and various host factors (1). Among the many types of cariogenic microorganisms in the human oral environment, *Streptococcus mutans* is the one that shows strong acid production, acid resistance, and biofilm formation capability

(2). Through the mechanism of adhesion to a solid surface, *S. mutans* is able to colonize the oral cavity and form bacterial biofilms. Biofilms are communities of microorganisms that grow attached to a surface or interphase and are embedded in a self-produced extracellular matrix (3). Biofilms are more resistant to adverse environmental conditions and antibiotics than planktonic cells (4), making them difficult to eliminate and control in biofilm-associated infectious diseases, such as dental caries. Biofilms serve as a cariogenic niche where cariogenic microorganisms colonize and produce acids, resulting in dental caries. *S. mutans* can decompose carbohydrates and produce glucans and acids, resulting in enamel demineralization and the progression of dental caries (5). *S. mutans* is the primary bacterium responsible for dental caries. It forms biofilms by synthesizing insoluble extracellular polysaccharide (EPS), catalyzed by streptococcal glucosyltransferases (Gtfs) (6). During conditions favoring the formation of dental caries, Gtfs secreted by *S. mutans* adsorb onto the enamel pellicle and catabolize sucrose to synthesize EPS. The EPS matrix serves as the biofilm structure, mediating tight adherence to the tooth enamel and bacteria. Lactate dehydrogenase, encoded by *ldh* gene, is one of the most important enzymes in the process of glycolysis, which generates lactic acid in *S. mutans*. Lactate dehydrogenase and lactic acid facilitate *S. mutans* to dissolve tooth minerals and cause dental caries (7). Therefore, it is necessary to implement anticariogenic strategies to inhibit the growth of *S. mutans* and prevent the formation of cariogenic biofilms.

There are various strategies used to prevent dental caries, the most common of which is regular tooth brushing and flossing. However, these mechanical procedures alone are inadequate in eliminating oral pathogens, including *S. mutans* (8). A more effective approach is to combine them with antimicrobial agents for regular dental hygiene, such as chlorhexidine (9) and antimicrobial peptides (10). These antimicrobial agents are able to reduce the population of cariogenic bacteria in biofilms. However, tooth discoloration is the most common side effect deterring patients from using chlorhexidine (11). The clinical application of antimicrobial peptides faces significant challenges, including susceptibility to proteolytic degradation, lack of information on potential *in vivo* toxicities, and comparatively high costs associated with peptide production (12, 13). Therefore, it is necessary to discover and apply more effective antimicrobial agents for the clinical prevention of dental caries.

Medicinal plants are a rich source of bioactive molecules with significant structural diversity. Plant-derived compounds offer a promising alternative due to their lower toxicity and higher selectivity (14). They have shown antimicrobial properties against a wide range of pathogens (15), some of which have already been used for dental caries prevention. For example, curcumin, an active compound in turmeric, has been reported to reduce the formation of *S. mutans* biofilm by inhibiting the activity of sortase A (16). Identifying natural compounds extracted from plants with anticariogenic activity and understanding their relevant mechanisms contribute to the development of novel antimicrobial agents for the clinical prevention of dental caries. Berberine (BBR), a bioactive alkaloid isolated from several botanicals, such as the Chinese herb *Coptis chinensis* Franch. and Berberis plants (17), has a wide range of pharmacological properties, including antimicrobial, antidiabetic, and anticancer activities (18). BBR has been found to have potential therapeutic effects on a wide range of diseases, including cardiovascular, metabolic (19), gastrointestinal (20), neurological diseases (21), and cancer (22). It has comparable therapeutic effects on type 2 diabetes, hyperlipidemia and hypertension with relatively low cost and without serious side effects (23). Although BBR has been extensively studied, research on its antimicrobial effect on oral infectious diseases is limited, and the mechanism of its antimicrobial activity on cariogenic oral bacteria is yet to be discovered. Given the biologically multifunctional nature of BBR, investigating its effect on oral pathogens would provide evidence for its potential clinical use.

This study investigated the antimicrobial effect of BBR on *S. mutans*, one of the most important cariogenic microorganisms, including the inhibitory effect on the planktonic bacteria growth, biofilm formation, and virulence factors. The membrane-destructive

bactericidal mechanism of BBR on *S. mutans* was elucidated, and the biocompatibility of BBR was also assessed by evaluating its effect on the cell viability of human oral cells and macrophages. This study provided evidence for the antimicrobial activity of BBR and contributed to the development of novel natural anticariogenic agents for cariogenic biofilm prevention and cariogenic virulence inhibition.

## RESULTS

### BBR inhibited *S. mutans* planktonic growth and biofilm formation

The MIC and MBC of BBR against planktonic *S. mutans* were 100 µg/mL assessed by *in vitro* test (Fig. 1A). The growth curve assay, which could show the proliferation of bacteria directly, demonstrated a dose-dependent inhibitory effect of BBR on the proliferation of *S. mutans* planktonic cells, with 100 µg/mL and 200 µg/mL of BBR completely inhibiting bacterial proliferation (Fig. 1B through D). BBR was found to inhibit the growth of *S. gordonii* DL-1 and *S. mitis* ATCC 6249 to some extent (Fig. 1E and F). Moreover, we treated *S. mutans* UA159 with BBR solutions for 5 and 10 min. The results of CFU counting assay

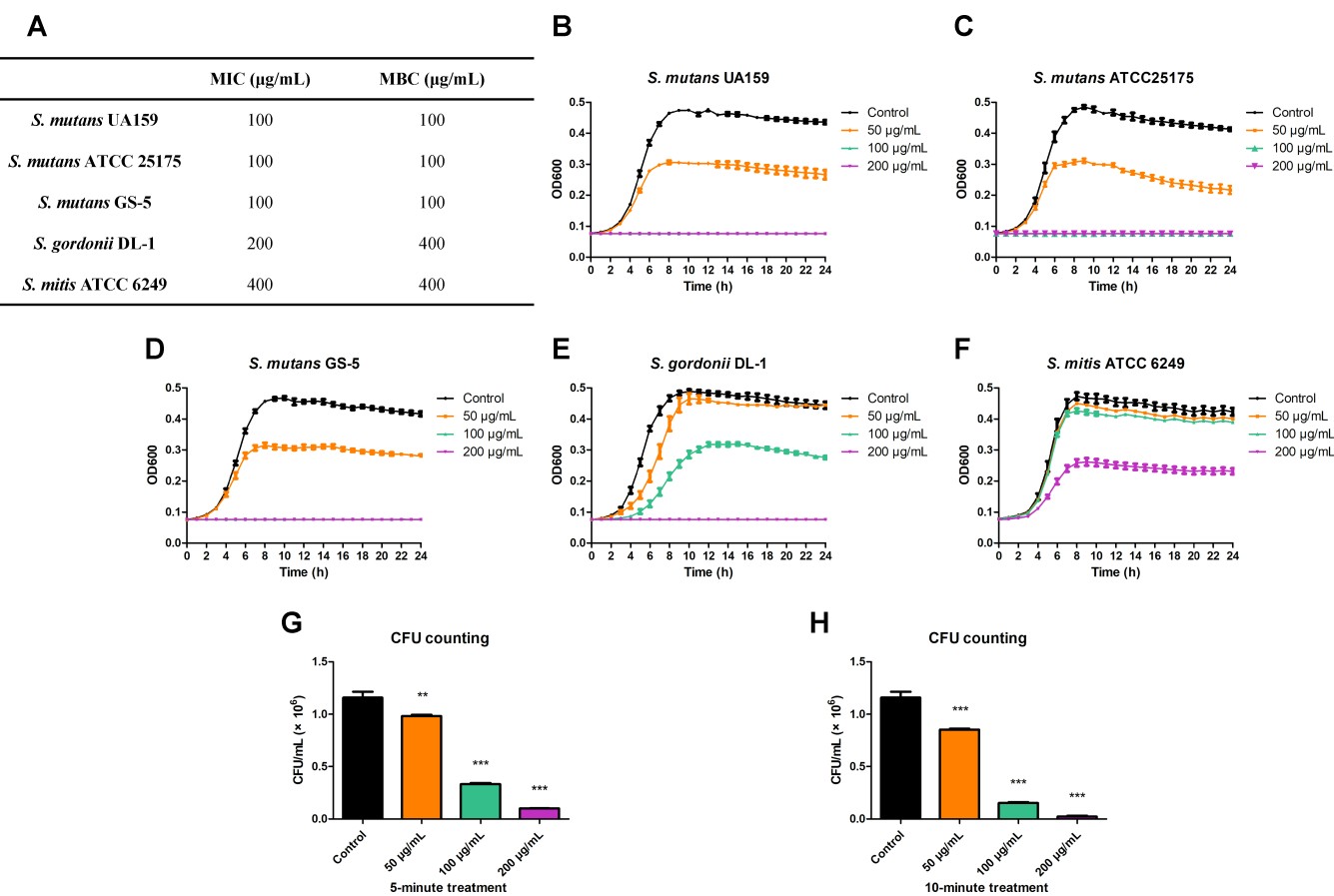

**FIG 1** Inhibitory effect of BBR on the growth of bacteria. (A) The MIC and MBC values of BBR against planktonic bacteria. $10^6$ CFU/mL bacteria in BHI broth were incubated with final concentrations of BBR ranging from 25 to 400 µg/mL anaerobically at 37°C for 24 h. The MIC was defined as the lowest concentration of BBR that inhibited visible bacterial growth. At the termination of the MIC assay, a volume of the culture was struck on BHIA and incubated to observe growth. The MBC was defined as the lowest concentration that yielded no colony growth by subculturing on BHIA plates. (B–F) The 24 h growth curve of planktonic *S. mutans* UA159, *S. mutans* ATCC 25175, *S. mutans* GS-5, *S. gordonii* DL-1, and *S. mitis* ATCC 6249 incubated under anaerobic conditions with/without treatment of BBR in a 96-well plate. The absorbance of each well was recorded every hour. (G and H) The short-term antibacterial effect of BBR on *S. mutans* UA159 was assessed by measuring the average number of CFU. *S. mutans* UA159 was treated with BBR for 5 min (G) and 10 min (H). The suspensions were then diluted and plated onto BHIA plates and incubated under anaerobic conditions at 37°C for 24 h to determine CFU counts. Values represent the means ± SD from three independent experiments (**$P < 0.01$, ***$P < 0.001$).

showed that 100 µg/mL and 200 µg/mL of BBR had comparable short-term antibacterial effect.

After that, we treated *S. mutans* UA159 with varying concentrations of BBR (200 µg/mL, 100 µg/mL, and 50 µg/mL) to evaluate the inhibitory effect of BBR on *S. mutans* biofilm formation. Crystal violet can stain the bacteria and EPS of the biofilm purple. Figure 2 shows the quantitative measurements of *S. mutans* biofilm formation. The results of the crystal violet staining assay showed that BBR treatment significantly reduced the total biomass of *S. mutans* biofilm, particularly in the 100 µg/mL and 200 µg/mL group (Fig. 2A and B). It is evident that treatment with BBR resulted in a significant reduction of biofilm accumulation of *S. mutans*, with 100 µg/mL and 200 µg/mL of BBR resulting in almost no biofilm formation on the culture plate. Moreover, the morphology of *S. mutans* biofilm under the treatment of BBR was examined by SEM (Fig. 2C). In the control group, *S. mutans* cells formed dense and multilayered biofilms. The bacteria were covered with large amounts of EPS (red arrows). The difference between the control group and 50 µg/mL group is not very significant under SEM. The application of 50 µg/mL of BBR resulted in a thin layer of biofilm with disrupted biofilm skeleton (yellow arrows) and enlarged biofilm pores (green arrows). In comparison to the control group, the biofilm formation of *S. mutans* was significantly inhibited by concentrations of 100 µg/mL and 200 µg/mL of BBR with only a small number of morphologically irregular cells and cell contents observed (purple arrows). The images suggest that BBR inhibits the biofilm formation of *S. mutans*.

## Quantification of *S. mutans* after treatment of BBR

The biofilm images formed with different concentrations of BBR after 24 h of incubation were observed using a CLSM. The stains SYTO 9/PI were used to evaluate membrane integrity and differentiate between membrane-intact and membrane-injured cells. Bacteria with intact cell membranes (live bacteria) were stained green (SYTO 9), while bacteria with damaged membranes (dead bacteria) were stained red (PI). An increase in red fluorescence was observed after the treatment with 50 µg/mL of BBR (Fig. 3A). The biofilm formation of *S. mutans* was completely inhibited by 100 µg/mL of BBR, with almost no bacteria adhered. No green fluorescence was observed, and faint red fluorescence was observed by CLSM, which was exhibited by few adhered dead bacteria and was difficult to observe in the figures. The Live/Dead ratio indicated effective elimination of BBR on *S. mutans*, with almost no live bacteria observed in the 100 µg/mL BBR-treated groups (Fig. 3B). Moreover, the CFU counting assay, the gold standard for enumerating viable bacteria, demonstrated that treatment with varying concentrations of BBR resulted in a significant reduction in CFU counts in the *S. mutans* biofilm (Fig. 3C). As for the biofilm eradication assay, *S. mutans* were cultured for 24 h to form mature biofilms. After that, biofilms were treated with varying concentrations of BBR solutions (200 µg/mL, 100 µg/mL, and 50 µg/mL) for 24 h to evaluate the biofilm eradicating activity of BBR. The result of CLSM showed the bactericidal effect of BBR. In the absence of BBR, a significant number of live bacteria were present in the biofilms (Fig. 4A). Following treatment with BBR on mature biofilms, the images of the BBR treatment groups showed more PI positive cells compared to the control group, indicating a higher number of dead bacterial cells. The results demonstrate that BBR reduces the number of live bacteria in *S. mutans* mature biofilm, indicating its effectiveness in eradicating the biofilm (Fig. 4B). The results of the CFU counting assay also confirm the inhibitory effect of BBR on *S. mutans* mature biofilm (Fig. 4C).

## BBR inhibited *S. mutans* virulence factors

The ability to synthesize EPS is one of the main virulence factors *S. mutans*. EPS aid in the permanent colonization of hard surfaces and in the development of the extracellular polymeric matrix *in situ* (24). CLSM, which could observe the three-dimensional images of the *S. mutans* biofilm, was used to quantify the amounts of bacteria and EPS synthesized by *S. mutans* under the treatment of BBR. The biofilms were cultured

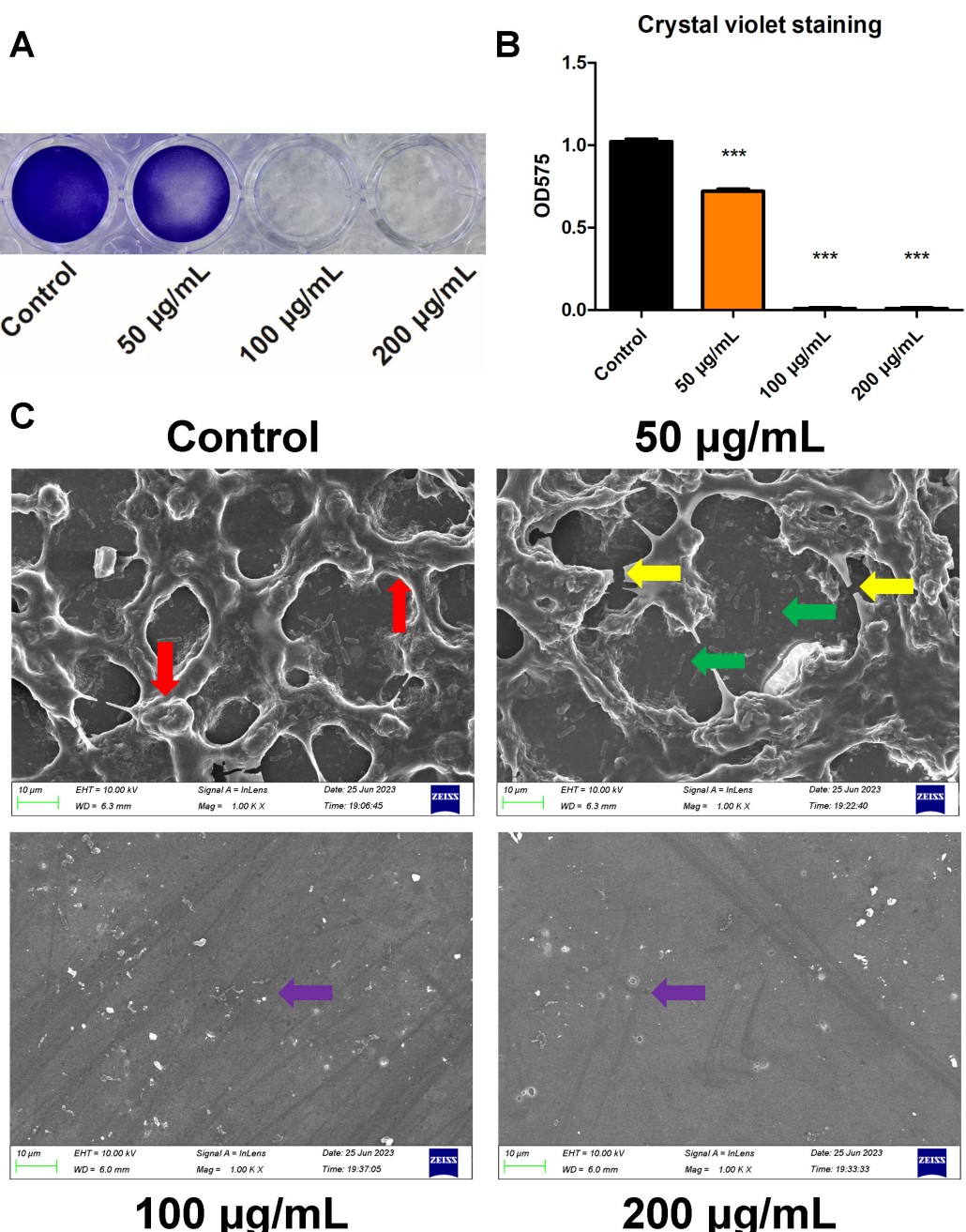

**FIG 2** Effect of BBR on the biofilm formation of *S. mutans*. In these assays, BBR was incubated with *S. mutans* in BHIS for 24 h to evaluate its effect on the biofilm formation. (A) The crystal violet staining of *S. mutans* 24 h biofilms under the treatment of BBR. The images were taken from a 24-well cell culture plate. (B) The results of the crystal violet staining assay applied to *S. mutans* biofilms after treated with different concentrations of BBR for 24 h in BHIS. (C) SEM micro-images of *S. mutans* 24 h biofilms on glass coverslips. In the control group, *S. mutans* bacteria formed dense biofilms covered with large amounts of EPS (red arrows). Disrupted biofilm skeleton (yellow arrows) and enlarged biofilm pores (green arrows) were observed under the treatment of 50 µg/mL of BBR. Only a small number of morphologically irregular *S. mutans* and cell contents were observed under the treatment of 100 µg/mL and 200 µg/mL of BBR (purple arrows). Values represent the means ± SD from three independent experiments (***$P < 0.001$).

on glass coverslips for efficient image capture and data collection. BBR was added immediately after bacterial dilutions were added to the culture plate, and the images were captured 24 h later. Figure 5A shows a series of three-dimensional micro-images

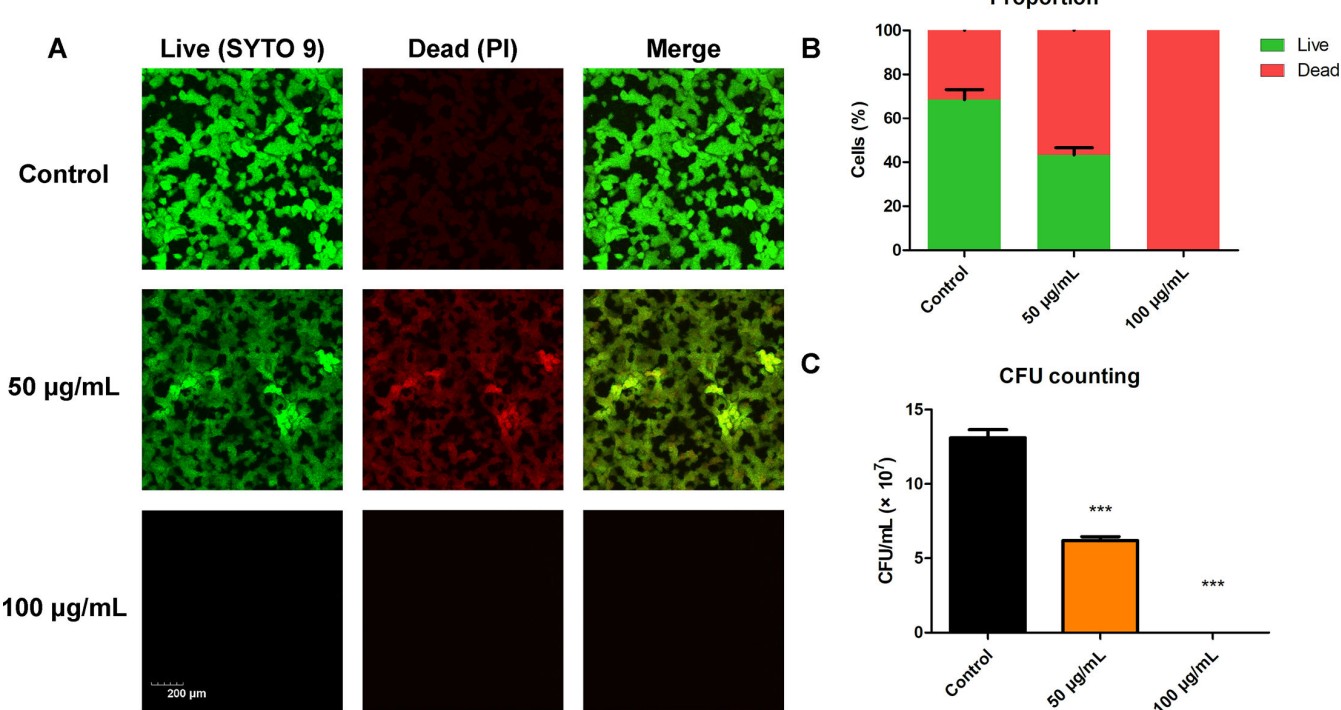

**FIG 3** Quantification of *S. mutans* biofilm after treatment of BBR. (A) Double-labeling imaging of *S. mutans* 24 h biofilms formed on glass coverslips. Live bacteria were green-labeled, and dead bacteria were red-labeled. (B) BBR reduced the proportion of live bacteria in *S. mutans* biofilms. (C) The effect of BBR on *S. mutans* biofilm formation was assessed by measuring the average number of CFU in the biofilm in one well of the culture plate. The analysis was performed using Image J COMSTAT software. Values represent the means ± SD from three independent experiments (***$P < 0.001$).

of *S. mutans* biofilms formed on glass coverslips, captured by CLSM. The bacteria and EPS were labeled green and red, respectively. Image J COMSTAT was used to analyze the distribution of bacteria and EPS in each layer of the *S. mutans* biofilms, as well as the relevant biomass. Moreover, the total biomass of bacteria and EPS in *S. mutans* biofilms was calculated according to the statistics. In the absence of BBR, the biofilm exhibited a uniform distribution with a relatively dense structure and complete surface coverage (Fig. 5B). Following treatment with BBR, the surface area covered by the biofilm decreased visibly, resulting in a significant reduction in biofilm biomass (Fig. 5C). The micro-images and statistical results demonstrate that bacterial proliferation and EPS synthesis were inhibited in the BBR-treated groups, resulting in a decrease in the accumulation of *S. mutans* adherent cells and EPS. Moreover, treatment with varying concentrations of BBR led to a significant reduction in the synthesis of water-insoluble EPS (Fig. 5D) in *S. mutans* biofilm, which is consistent with the result of SEM. Acidogenicity is another main virulence factor of *S. mutans*. The impact of BBR on acid production by *S. mutans* was evaluated using the lactic acid assay and glycolytic pH drop assay. *S. mutans* was cultured in the presence of varying concentrations of BBR, and the production of lactic acid was measured. Figure 5E showed that BBR decreased the lactic acid production of the biofilms compared to the control group. However, acid production does not inevitably lead to dental erosion. The critical pH of dental enamel is approximately 5.5 and any solution with a lower pH value may cause erosion (25). Therefore, the glycolytic pH drop assay was further conducted. As shown in Fig. 5F and G, the pH values of the BBR treated group significantly differed from those of the control group. In general, BBR reduced the rate of pH drop compared to the control group. In 1% glucose-supplemented group, the final pH values decreased from 7.22 ± 0.02 to 4.98 ± 0.09 in the control group after 270 min of incubation. Treatment with BBR at concentrations of 50 µg/mL, 100 µg/mL, and 200 µg/mL increased the terminal

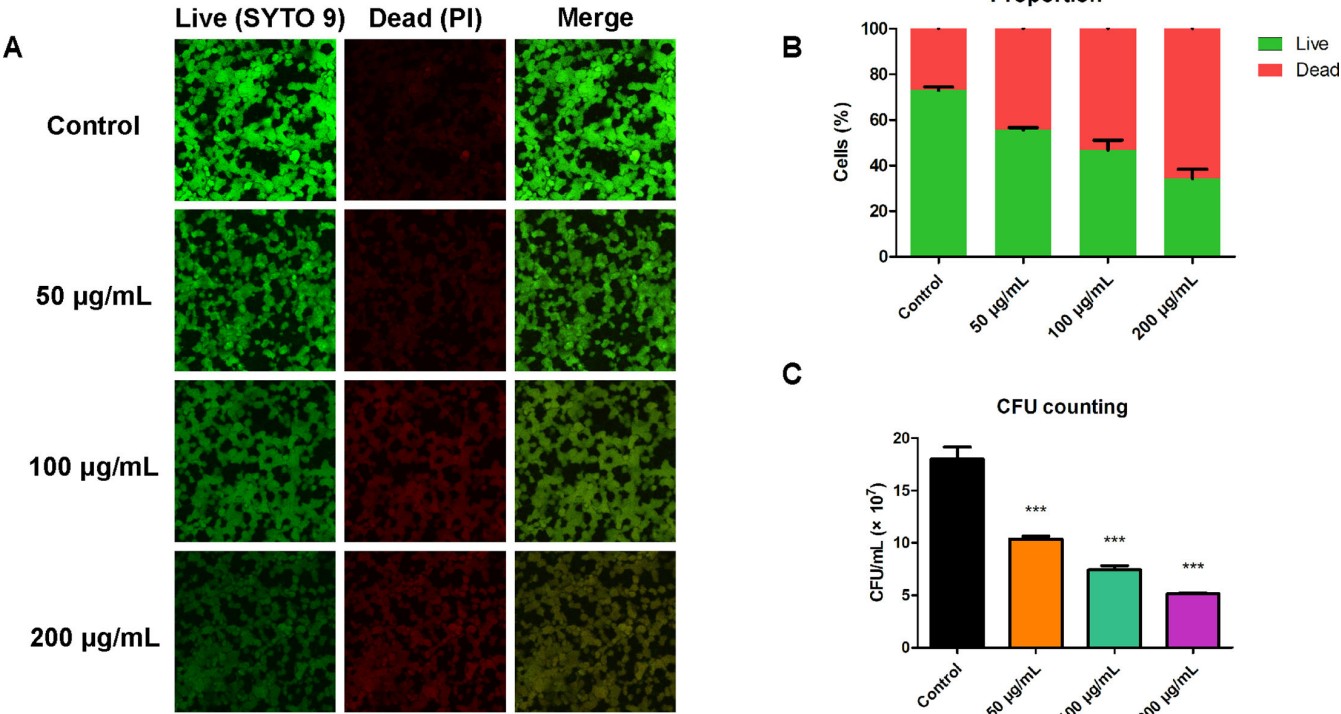

**FIG 4** Effect of BBR on eradicating the *S. mutans* mature biofilms. (A) Double-labeling imaging of *S. mutans* mature biofilms treated with BBR. Live and dead bacteria were green-labeled and red-labeled, respectively. (B) BBR decreases the proportion of live bacteria in *S. mutans* mature biofilms. (C) The effect of BBR on eradicating the *S. mutans* mature biofilms was measured by the determination of the average number of CFUs in the biofilm. The *S. mutans* biofilms were cultured for 24 h without BBR. Then, BBR solution was added to each well for another 24 h. The data were collected at 48 h. The analysis was performed using Image J COMSTAT software. Values represent the means ± SD from three independent experiments (***$P < 0.001$).

pH to 5.97 ± 0.02, 6.74 ± 0.01, and 6.86 ± 0.01, respectively. The results showed that the control group experienced the greatest decrease in pH within the first 30 min of incubation, dropping from 7.22 ± 0.02 to 6.34 ± 0.05. In contrast, the group treated with 200 µg/mL BBR experienced the smallest drop in pH, from 6.86 ± 0.01 to 7.11 ± 0.02 (Fig. 5F). The pH reduction in 0.1% glucose-supplemented group is less pronounced overall, particularly showing a significantly attenuated initial drop compared to the 1% glucose-supplemented group (Fig. 5G). The results indicated that BBR inhibits *S. mutans* acidogenicity and alleviates pH drop.

## The bactericidal mechanism of BBR on *S. mutans*

Aggregation, which occurs due to chemical or electrostatic interaction between cell surface molecules, can contribute to bacterial adherence and biofilm formation (26). The adhesion of *S. mutans*, which includes a sucrose-independent mode, is one of the most important steps in establishment of dental caries, and its inhibition could be a strong step against the impairment of its virulence (27). As shown in Fig. 6A, the aggregation rate of *S. mutans* reached 17.41% ± 0.1379% after 6 h incubation. A dose-dependent decrease in bacterial aggregation was observed with different concentrations of BBR. The results indicated that BBR inhibited the aggregation of planktonic bacteria, thereby inhibiting the formation of biofilm. Depolarization of bacterial membranes was evaluated by DiSC3(5), a kind of membrane potential sensitive dye, which concentrates within the lipid bilayer resulting in the dye self-quenching (28–31). The potential dissipates if when membranes are depolarized and perturbed, resulting in the release of DiSC3(5) into the solution and an increase in fluorescence intensity (32), which indicates that the cell membranes are targeted. The concentration of 100 µg/mL and 200 µg/mL of BBR had stronger effects on depolarizing bacterial cytomembranes than

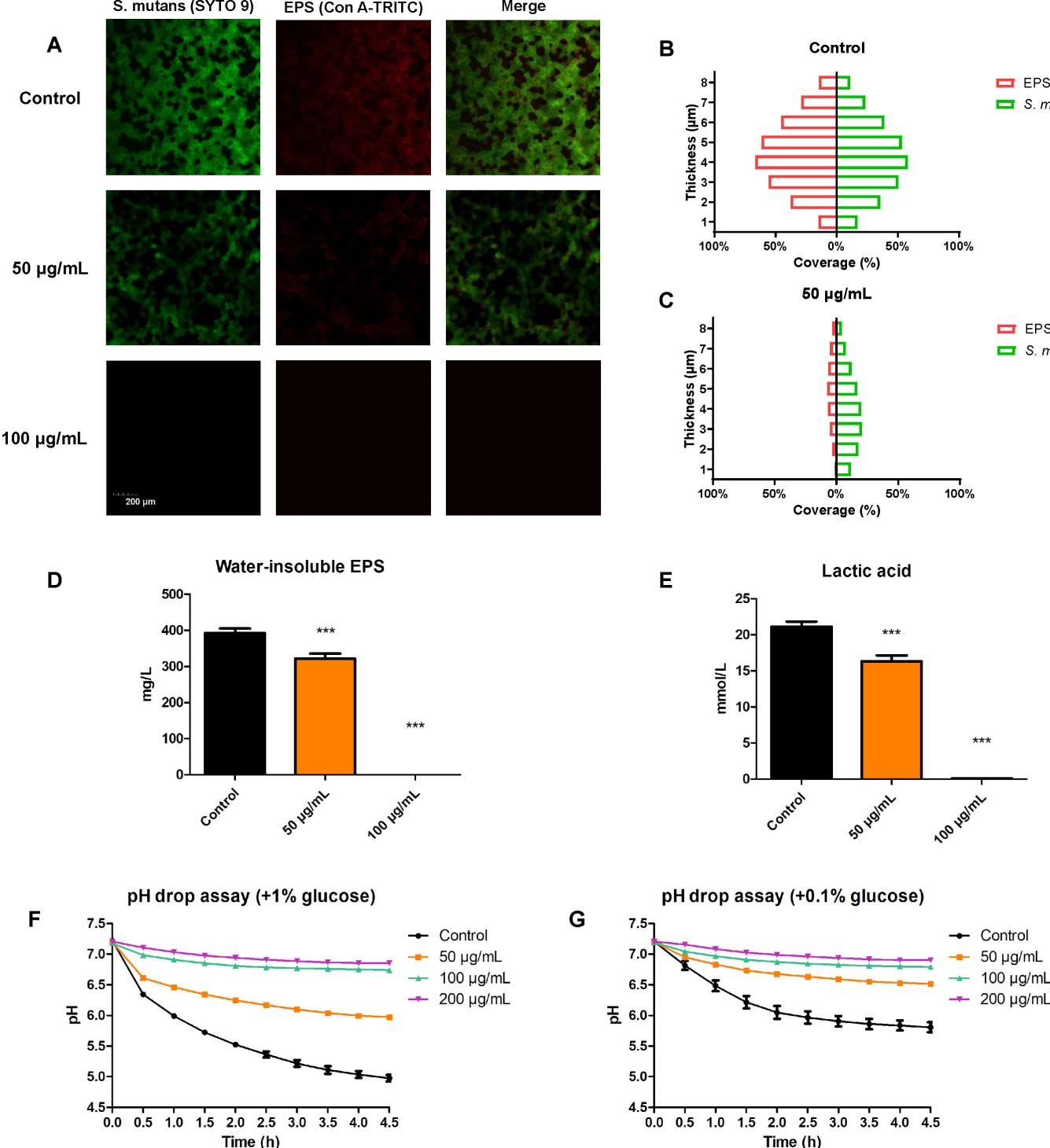

**FIG 5** BBR inhibited *S. mutans* virulence factors. (A) Effect of BBR on the biofilm structure of *S. mutans* observed by CLSM. Double-labeling imaging of *S. mutans* 24 h biofilm formed on glass coverslips. The fluorescence (SYTO 9) marks the live bacteria, while the red fluorescence (Concanavalin A-TRITC) marks the EPS synthesized by *S. mutans*. (B) Quantification of the amounts of EPS and bacteria in each scanned layer of *S. mutans* 24 h biofilm without the treatment of BBR. (C) Quantification of the amounts of EPS and bacteria in each scanned layer of *S. mutans* 24 h biofilm with the treatment of 50 µg/mL BBR. (D) Quantitative measurement of water-insoluble EPS by anthrone-sulfuric method. (E) Measurement of lactic acid production. (F and G) Effect of BBR on *S. mutans* glycolytic pH drop under 1% (F) and 0.1% (G) glucose. Values represent the means ± SD from three independent experiments (***$P < 0.001$).

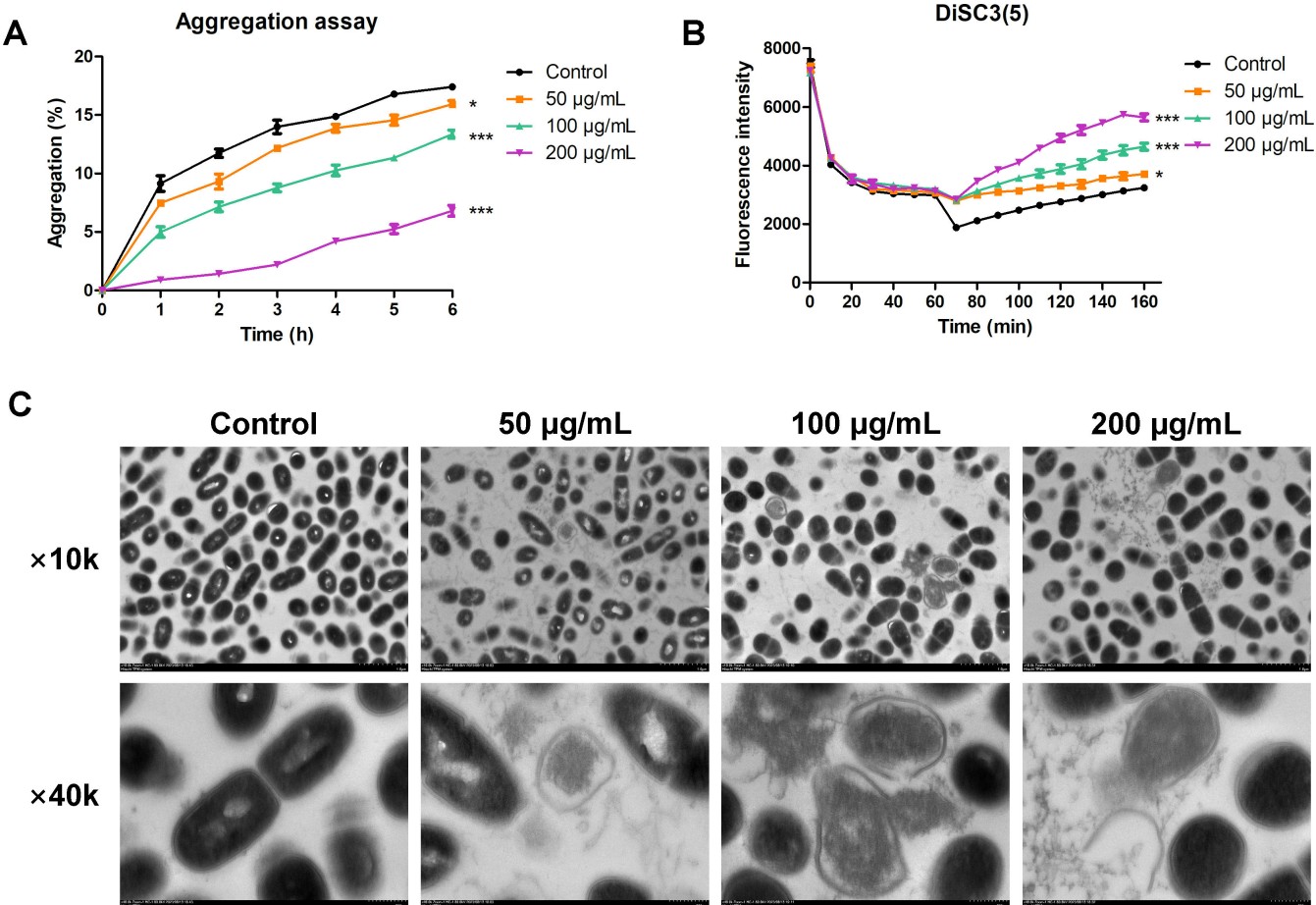

**FIG 6** The bactericidal mechanism of BBR on *S. mutans*. (A) Effect of BBR on aggregation. (B) Membrane depolarization activity of BBR was tested using DiSC3(5). (C) TEM micrographs of *S. mutans*. Values represent the means ± SD from three independent experiments (*$P < 0.05$, ***$P < 0.001$).

the concentration of 50 µg/mL of BBR or no BBR (Fig. 6B), indicating the intensified perturbing ability of BBR on the membrane of *S. mutans* at the concentration of 100 µg/mL and 200 µg/mL. TEM was used to observe the effect of BBR on the morphology of *S. mutans*. In the control group, the morphology of *S. mutans* was uniform and smooth with intact cell membrane. The TEM images of *S. mutans* showed membrane damage caused by BBR. The treatment of BBR altered the membrane structures of *S. mutans*, resulting in bacterial lysis and the release of cell contents (Fig. 6C). The results demonstrated that BBR acted as an antimicrobial agent by disrupting the membrane integrity of *S. mutans*.

## BBR inhibited *S. mutans* virulence genes expression

To investigate the mechanism of BBR on the adhesion and biofilm formation of *S. mutans*, as well as its inhibitory effects at the transcriptional level beyond direct bactericidal activity, the relative expression levels of characteristic virulence genes were investigated by qRT-PCR on *S. mutans* treated with BBR. The formation of *S. mutans* biofilm is closely related to the activity of Gtf proteins. The reduced expression of Gtf-encoding genes (*gtfB*, *gtfC*, and *gtfD*) may disrupt the formation and integrity of *S. mutans* biofilm (33, 34). Therefore, the relative expression levels of *gtfB*, *gtfC*, and *gtfD* were investigated. Compared to the control groups, the expression of *gtfD* was downregulated in the BBR-treated groups. Moreover, the expression of *gtfB* and *gtfC* was downregulated in the 100 µg/mL and 200 µg/mL groups (Fig. 7), indicating the inhibitory effect of BBR on the expression of gtf genes. In addition, *ldh* gene encodes lactate dehydrogenase,

generating lactic acid in *S. mutans*. Three tested genes were found to be related to the two-component signal transduction system and regulate the bacterial biofilm formation, known as *vicR*, *liaR*, and *comD*. Decreased expression of these virulence genes in BBR-treated groups compared with the control group was also revealed by qRT-PCR.

## BBR exhibited good biocompatibility

In order to evaluate the biocompatibility of BBR in the oral cavity, we treated human gingival epithelial cells (HGECs), human oral keratinocytes (HOKs), human periodontal ligament cells (HPDLCs), human gingival fibroblasts (HGFs), RAW 264.7, and THP-1 with varying concentrations of BBR for 0.5 h and 8 h to evaluate its short- and long-term effects and measured cell viability using a CCK-8 assay (Fig. 8). HGECs and HOKs represented the oral mucosa which is in direct contact with the BBR, while HPDLCs, HGFs, RAW 264.7, and THP-1 which is not in direct contact with BBR. When treated with BBR for 0.5 h, none of the tested concentrations of BBR exhibited a significant inhibitory effect on the cell viability of HOKs, HGFs, RAW 264.7, or THP-1. However, the concentration of 200 µg/mL of BBR slightly decreased the cell viability of HGECs and HPDLCs. Moreover, 200 µg/mL of BBR inhibited the cell viability of HGECs, HOKs, HPDLCs, HGFs, RAW 264.7, and THP-1 when treated for 8 h, while 50 µg/mL and 100 µg/mL of BBR exhibited not statistically significant effect. This indicates that BBR has good biocompatibility when applied to cultured cells to a certain extent.

## DISCUSSION

Biguanides were discovered in *Galega Officinalis*, a plant that has been used in folk medicine for centuries and has developed into the commonly used hypoglycemic agent Metformin (35). Tu found that *Artemisia annua* L. could alleviate malaria symptoms in Ge Hong's A Handbook of Prescriptions for Emergencies, an ancient Chinese medicine work, and effectively reduce the mortality rate of malaria (36). It is evident that traditional medicinal plants are a valuable resource that worth exploring. *Coptis chinensis* Franch. is a commonly used and precious Chinese herb that contains various bioactive alkaloids, including BBR, coptisine, and epiberberine. Coptisine could protect against hyperuricemia-induced renal inflammatory damage, oxidative stress, and mitochondrial

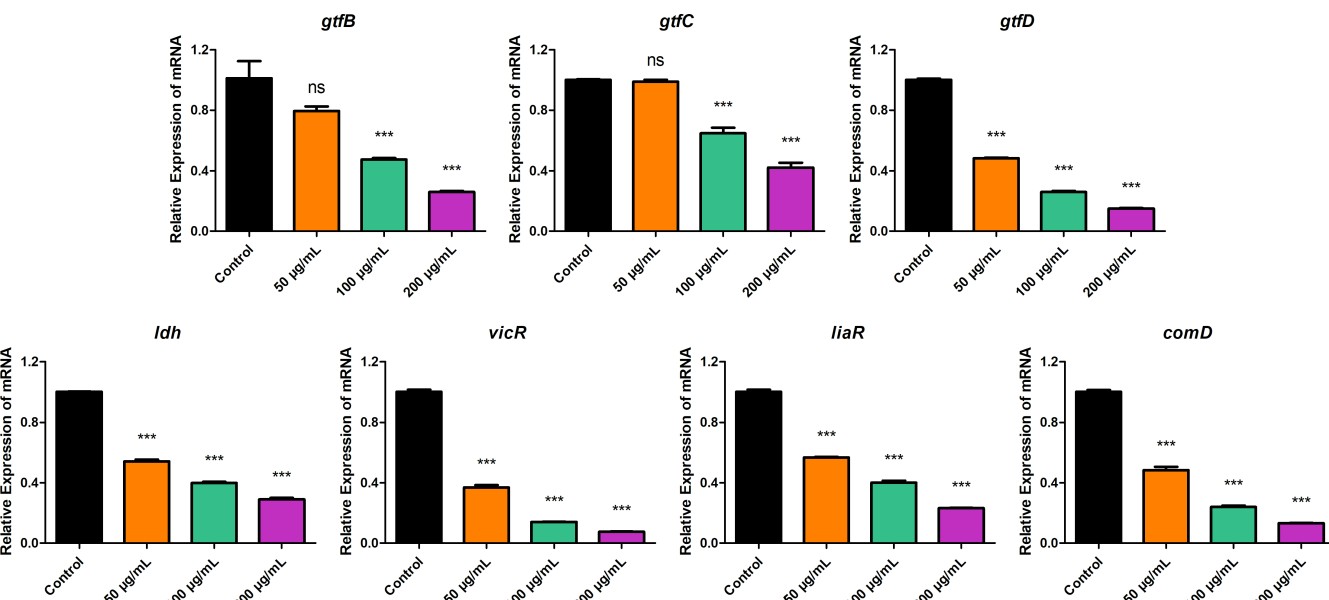

**FIG 7** Effects of BBR on *S. mutans* gene expression. The relative mRNA expressions of *gtfB*, *gtfC*, *gtfD*, *ldh*, *vicR*, *liaR*, and *comD* were measured by qRT-PCR. *S. mutans* UA159 16S rRNA was used as an internal control. Values represent the means ± SD from three independent experiments (***$P < 0.001$, ns, not statistically significant compared to the untreated control group).

# Treated with BBR for 0.5 h

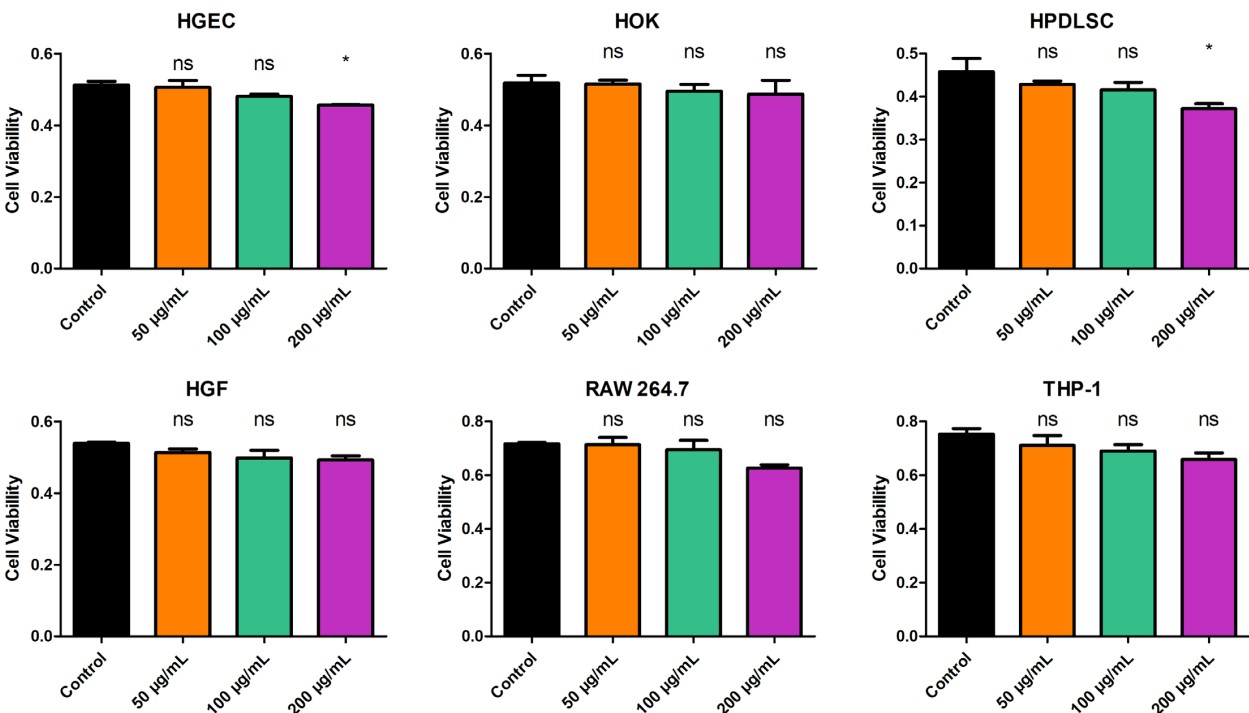

# Treated with BBR for 8 h

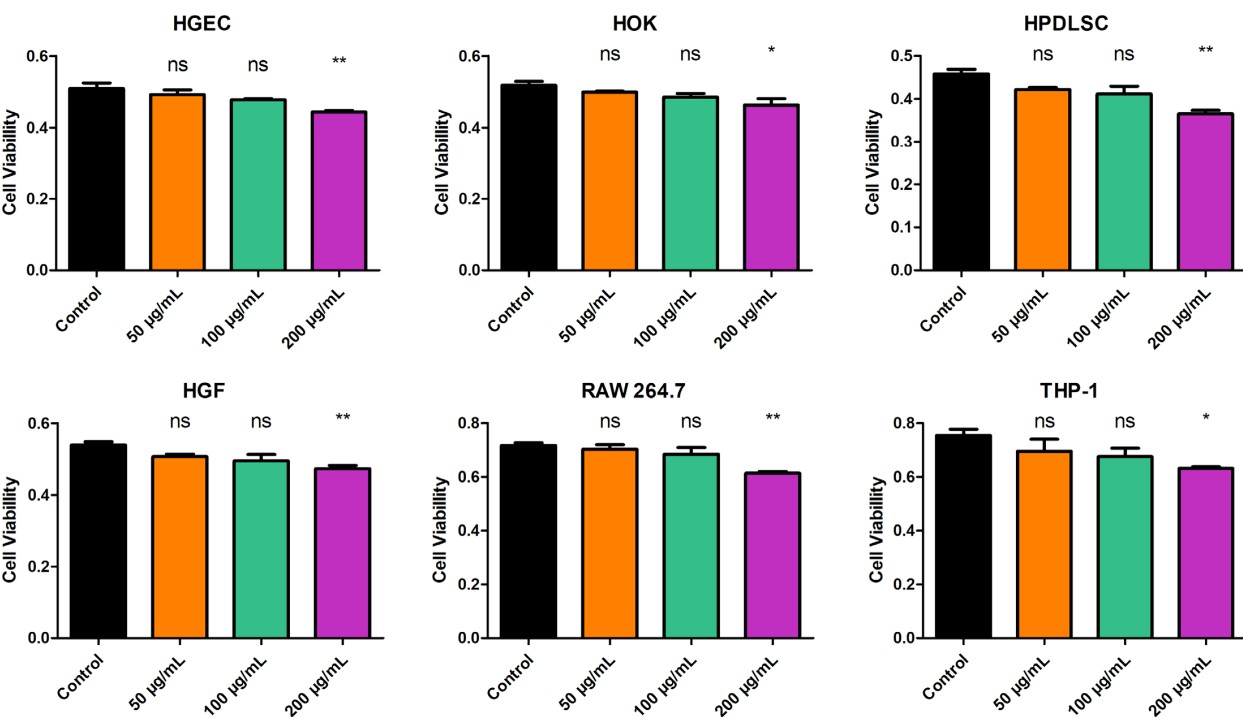

**FIG 8** Biocompatibility of BBR evaluated by CCK-8 assay. The assay measured the effect of BBR on the cell viability of HGECs, HOKs, HPDLCs, HGFs, RAW 264.7, and THP-1. The values represent the means ± SD from three independent experiments (*$P < 0.05$, ns, not statistically significant compared to the untreated control group).

apoptosis (37). Epiberberine could suppress hepatic triglyceride synthesis, ameliorate liver steatosis, and improve the gut microbiome (38). BBR, a bioactive alkaloid isolated from the Chinese herb *Coptis chinensis* Franch., is a traditional anti-infective drug that has been confirmed to have antibacterial activity against a variety of bacteria. BBR has been found to effectively inhibit *Salmonella Typhimurium* (39) and *Staphylococcus epidermidis* (40), indicating its antimicrobial activity. This study aimed to investigate the effects of BBR on the planktonic cells, biofilm formation, mature biofilm, virulence factors, gene expression of *S. mutans* and to elucidate the potential mechanisms involved.

This study conducted preliminary *in vitro* experiments to investigate the anti-caries effects of BBR. The results showed that BBR exhibited inhibitory effects on *S. mutans* and its biofilms. The MIC of the extract of *Cedrus deodara* (41) on *S. mutans* is 6250 µg/mL, cinnamaldehyde (7) is 1,000 µg/mL, and resveratrol (42) is 800 µg/mL. In this study, the MIC of BBR on *S. mutans* was found to be 100 µg/mL (Fig. 1A). This value is lower than that of the natural products in previous studies, indicating that BBR exhibits favorable inhibitory effects on *S. mutans*.

Biofilms play a significant role in the development of dental caries, by providing a cariogenic microenvironment that resists external influences and enables *S. mutans* to exhibit cariogenic toxicity continuously. The study shows that BBR can inhibit the formation of biofilms. Compared to the control group, the results of the crystal violet staining assay indicated that the biofilm of *S. mutans* was more transparent under the treatment of 50 µg/mL of BBR (Fig. 2A). Furthermore, the quantitative evaluation also confirmed a decrease in biomass (Fig. 2B). Moreover, CLSM was used to assess the live and dead bacterial counts in *S. mutans* biofilms. The results demonstrated that treatment with 50 µg/mL of BBR resulted in a decrease in the number of live bacteria and an increase in the number of dead bacteria (Fig. 3). The biofilm formation of *S. mutans* was completely inhibited by 100 µg/mL of BBR, with almost no bacteria adhered. No green fluorescence was observed, and faint red fluorescence was observed by CLSM, which was exhibited by few adhered dead bacteria and was difficult to observe in the figures.

The study demonstrated that BBR exhibited inhibitory effects on planktonic *S. mutans* and its biofilm formation. However, the structure of mature biofilm can help resist adverse external influences. Even antibacterial agents with good antibacterial effects against planktonic bacteria have uncertain anti-biofilm effects. For instance, ceftazidime has notable antibacterial effects against planktonic *Pseudomonas aeruginosa*, yet it is challenging for ceftazidime to penetrate into the biofilm of *Pseudomonas aeruginosa*, resulting in unfavorable anti-biofilm effects (43). Therefore, the mature biofilm of *S. mutans* was formed, and the anti-biofilm effect of BBR on *S. mutans* mature biofilm was investigated. The results of CLSM demonstrated that BBR exhibited antibacterial effects against *S. mutans* in mature biofilm, which suggested that BBR could penetrate into the biofilm and kill embedded bacteria. The penetration effect beyond the protection of biofilm should be further evaluated. With the increase in the concentration of BBR, the number and proportion of live bacteria in the mature biofilm gradually decreased (Fig. 4). Compared to the previous results, 100 µg/mL and 200 µg/mL of BBR were found to be effective in inhibiting the biofilm formation of *S. mutans*. However, they were unable to eradicate the bacteria in the mature biofilm within the experimental timeframe, suggesting that the mature biofilm can help resist the effect of BBR.

The microscopy methods used in this study clearly showed the effect of BBR on *S. mutans* biofilm. In the control group, *S. mutans* bacteria formed dense biofilms covered with large amounts of EPS (Fig. 2C, red arrows). Only a small number of morphologically irregular *S. mutans* and cell contents were observed under the treatment of 100 µg/mL and 200 µg/mL of BBR (purple arrows). Disrupted biofilm skeleton (yellow arrows) and enlarged biofilm pores (green arrows) were observed, potentially due to the reduced synthesis of EPS under the treatment of 50 µg/mL of BBR. The difference between the control group and 50 µg/mL group is not very significant under SEM, which is a qualitative technique for observing biofilm details. Therefore, CLSM was further used to quantify bacterial counts and EPS in *S. mutans* biofilms. In the control group, the

surface of the glass coverslips was covered by dense and multilayered biofilms with substantial amounts of EPS. Treatment with 50 µg/mL of BBR resulted in a decrease in the amounts of live bacteria and EPS (Fig. 5A through C), resulting in the formation of an incomplete structured biofilm. The results of microscopy assays suggested that BBR inhibited *S. mutans* proliferation and EPS synthesis. It has been largely accepted that the ability to synthesize large quantities of EPS from sucrose and the ability to transport and metabolize a wide range of carbohydrates into organic acids are considered the principal cariogenic virulence of *S. mutans* (24). Therefore, the effect of BBR on the synthesis of EPS and the production of organic acid is important to evaluate the virulence factors of *S. mutans*. BBR can inhibit the formation of EPS (Fig. 5D), reducing the amount of matrix and the strength of biofilms, making it easier to clear the biofilms. Moreover, BBR can inhibit acid production (Fig. 5E) and pH decrease (Fig. 5F and G), which helps alleviate acid erosion on dental enamel.

Previous research has primarily focused on the effectiveness of natural products in preventing dental caries. This study delves deeper by conducting preliminary experiments on the anti-caries mechanism of BBR. A voltage-sensitive DiSC3(5) probe was used to investigate the effect of BBR on the membrane of *S. mutans*. The damaged bacteria exhibited depolarized membranes, resulting in the release of the probe and the increase in the fluorescence. An increase in the fluorescence intensity was observed, indicating a change in the membrane potential of *S. mutans* following BBR treatment (Fig. 6B), thereby reinforcing our hypothesis on the mechanism of BBR. The fluorescence intensity of *S. mutans* treated with DiSC3(5) revealed that BBR can depolarize and perturb the bacterial cytomembranes of *S. mutans*, causing a decrease in *S. mutans* aggregation (Fig. 6A). This was further substantiated by the analysis of morphological changes conducted via TEM. BBR was found to damage the cell membrane of *S. mutans*, resulting in bacterial lysis and the release of cell contents (Fig. 6C). The results demonstrated that BBR treatment altered the membrane potential of *S. mutans*, resulting in bacterial damage and a reduction in *S. mutans* aggregation. Similar results were observed for *Staphylococcus aureus* (44) and *Candida albicans* (45), where BBR induced damage to the cell membrane and consequently resulted in the release of intracellular contents, which is consistent with our findings. The results suggested that BBR exhibited inhibitory effects on *S. mutans* by decreasing the aggregation and damaging the cell membrane of *S. mutans*.

Downregulation of virulence genes might explain the inhibitory effects of BBR on various cariogenic properties. *S. mutans* is a key contributor to the formation of the EPS matrix in dental biofilms. The EPS, which are mostly glucans synthesized by Gtfs, provide binding sites that promote accumulation of microorganisms on the tooth surface and further establishment of pathogenic biofilms (46). *S. mutans* expresses three genetically distinct Gtfs, each of which appears to play a different but overlapping role in the formation of virulent plaque. GtfC is adsorbed onto the enamel within the pellicle, while GtfB binds avidly to bacteria, promoting tight cell clustering and enhancing plaque cohesion. GtfD forms a soluble, readily metabolizable polysaccharide and acts as a primer for GtfB (47). Therefore, the relative mRNA expression levels of *gtfB*, *gtfC*, and *gtfD* were measured by qRT-PCR. The study revealed that the expression levels of gtf genes were downregulated when treated with BBR. The two-component signal transduction system of *S. mutans*, including VicRK, LiaSR, and ComDE, plays an important role in bacterial environmental adaptation, production of virulence factors, self-defense, and biofilm formation (48). The VicRK system was found to modulate sucrose-dependent adhesion, biofilm formation, genetic competence development, and regulate the expression of several virulence-associated genes affecting synthesis and adhesion to polysaccharides in *S. mutans*, including *gtfB*, *gtfC*, and *gtfD* (49). According to the results, the expression level of *vicR* gene was inhibited by BBR, suggesting the potential mechanism of the decreased synthesis of Gtfs. The LiaSR system has been implicated in several functions in *S. mutans*, including cell division, acid tolerance, biofilm formation, and antibiotic resistance (50). In this study, the decreased expression level

of *liaR* gene under the effect of BBR may result in the decreased biofilm formation of *S. mutans*. A signal peptide-mediated quorum-sensing system encoded by ComDE has been found to play a central role in regulation of genetic competence, bacteriocin production, biofilm formation, and stress response (51). The repression of this gene would attenuate internal communication quorum-sensing mechanism in *S. mutans* and further inhibit biofilm formation. The results of this study showed that BBR inhibited the ComDE system, leading to the inhibition of biofilm formation. According to the results, the decreased expression of *vicR*, *liaR*, and *comD* may lead to the decreased biofilm formation of *S. mutans*. In addition, the decreased expression level of *ldh* gene, encoding lactate dehydrogenase that generate lactic acid in *S. mutans*, suggest the potential mechanism of the inhibition of acid production, which is consistent with the results of lactic acid assay and glycolytic pH drop assay. In conclusion, in addition to its direct antibacterial effects, BBR also suppressed the expression of virulence genes in *S. mutans*, suggesting that the anti-caries mechanism of BBR involved both bactericidal and virulence-inhibiting effects.

On one hand, the results of this study indicated that the concentration of 100 µg/mL of BBR is MIC for *S. mutans*. The treatment with 100 µg/mL of BBR was found to inhibit the biofilm formation of *S. mutans*. This is consistent with the results of qRT-PCR assay, which demonstrated that this concentration of BBR could inhibit the expression of gtf genes, thereby inhibiting the synthesis of EPS and the formation of biofilm. On the other hand, the results of this study indicated that the treatment with 50 µg/mL of BBR inhibited the proliferation of *S. mutans*, with dead bacteria observed by CLSM. The mechanism of the inhibitory effect of 50 µg/mL of BBR was found to be the depolarization and damage to the *S. mutans* cell membrane (Fig. 6). Consequently, PI staining solution was observed to enter the bacteria and exhibit red fluorescence when observed by CLSM (Fig. 3A). Furthermore, the depolarization and damage of the *S. mutans* cell membrane contributed to the inhibition of *S. mutans* proliferation and biofilm formation. Previous research has found that decreased expression of gtf genes results in the formation of less biofilm (52). Glucans, synthesized by *GtfB* and *GtfC*, are essential for creating a matrix that promotes the co-aggregation of bacterial cells (53). The treatment with 50 µg/mL of BBR resulted in a slight reduction in the expression of *gtfB* and *gtfC* although the results were not statistically significant when compared to the control group. This is consistent with the results of the aggregation assay, which demonstrated that this concentration of BBR could slightly inhibit the aggregation of *S. mutans*. GtfD forms a soluble, readily metabolizable polysaccharide (47), which is associated with the metabolism and proliferation of *S. mutans*. The treatment with 50 µg/mL of BBR could reduce the expression of *gtfD* (Fig. 7). The result is consistent with the result of the growth curve assay, which demonstrated that 50 µg/mL of BBR could inhibit the proliferation of *S. mutans* (Fig. 1B). This resulted in the inhibition of biofilm formation, which manifested as a more transparent and incomplete structured biofilm (Fig. 2) covered with less amounts of EPS (Fig. 5). Although 50 µg/mL of BBR did not exhibit strong antibacterial effects on *S. mutans*, it exhibited good inhibitory effects on the virulence factors of *S. mutans*. The treatment with 50 µg/mL of BBR inhibited the synthesis of EPS (Fig. 5D), production of organic acids (Fig. 5E), and drop of glycolytic pH curve (Fig. 5F and G), thereby inhibiting the progression of dental caries.

When evaluating a new agent for clinical use, it is important to consider its biocompatibility. In this study, the biocompatibility of BBR was evaluated by measuring its effect on the cell viability of HGECs, HOKs, HPDLCs, HGFs, RAW 264.7, and THP-1. HGECs and HOKs represented the oral mucosa in direct contact with BBR, while HPDLCs, HGFs, RAW 264.7, and THP-1 represented the submucosal tissues. The results (Fig. 8) showed that treatment with BBR for 0.5 h did not significantly affect the cell viability of HOKs, HGFs, RAW 264.7, or THP-1, while a slight inhibition of cell viability was observed in HGECs and HPDLCs at a concentration of 200 µg/mL of BBR. Moreover, 200 µg/mL of BBR inhibited the cell viability of HGECs, HOKs, HPDLCs, HGFs, RAW 264.7, and THP-1 when treated for 8 h, while 50 µg/mL and 100 µg/mL of BBR exhibited not statistically significant effect,

indicating that long exposure to high concentration of BBR would affect the cell viability. The antibacterial effects and cytotoxicity of BBR show consistency. While 200 µg/mL of BBR serves as a higher concentration with enhanced antibacterial effects, it also increases cytotoxicity, highlighting the importance of concentration selection for antibacterial agents. Given that 100 µg/mL of BBR exhibited comparable short-term antibacterial effect (Fig. 1G and H) and favorable anti-caries effects with not statistically significant effect on the viability of HGECs, HOKs, HPDLCs, HGFs, RAW 264.7, and THP-1, BBR is a biocompatible and suitable option for clinical use as a novel anticaries agent.

BBR has been found to have potential therapeutic effects on a wide range of diseases, including cardiovascular, metabolic (19), gastrointestinal (20), neurological diseases (21), cancer (22), and have inhibitory effects on the growth and biofilm formation of *S. mutans* and *Streptococcus sobrinus* (54, 55). This study not only confirmed the antibacterial and anti-biofilm effects of BBR reported in previous studies but also further investigated its effects on virulence factors and cell viability, as well as its antibacterial mechanism, thereby enriching the investigation on BBR. In this study, BBR showed effective antimicrobial activity against *S. mutans* and remarkable biocompatibility. The sub-MIC of BBR exhibited inhibitory effects on the proliferation, biofilm formation, and virulence factors of *S. mutans*, whereas the MIC of BBR exhibited not only killing effects on *S. mutans* but also inhibitory effects on the virulence factors. The use of natural compounds is becoming increasingly important in the discovery of novel antimicrobial agents, particularly in light of the growing number of multidrug-resistant bacterial strains and increasing antibiotic resistance. This highlights the need for further research into the clinical applications of novel antimicrobial natural compounds. The study has several limitations. First, the study focuses on the *in vitro* effects of BBR against *S. mutans*, which may not accurately reflect the *in vivo* effects of BBR in clinical application due to the complex interaction of oral microbiota and the influence of oral environment. While *S. mutans* seem to be sensitive to BBR, future studies should further investigate the effects of BBR against other bacteria and multi-strain biofilms to clarify the mechanism of BBR more comprehensively. Second, considering the favorable antimicrobial effect of BBR on *S. mutans*, BBR is expected to play an anticaries role as a kind of gargle or toothpaste additive, which requires further research to investigate the minimum effective time and long-term biosafety *in vivo*. In addition, an effective concentration of BBR, in the presence of resveratrol, could be decreased even to 50% in cancer treatment (56), which suggests that the application of BBR in synergy with other agents may enhance the effects of BBR. Therefore, strategies to enhance the antimicrobial efficacy of BBR, including the combination of BBR with other antimicrobial agents or natural compounds to enhance its anticaries effects, would contribute to the clinical application of BBR. Notwithstanding these limitations, the study has important implications for the potential clinical application of BBR in the prevention of dental caries and may contribute to the development of novel anticaries approaches.

## MATERIALS AND METHODS

### Chemicals, bacterial strain, and growth conditions

Berberine hydrochloride (hereafter referred to as berberine, BBR) used in this study was obtained from Macklin Inc. (B796571, ≥ 99%). BBR was dissolved in double distilled water and filtered before used. The bacterial strain *S. mutans* UA159 was obtained from the American Type Culture Collection (ATCC). The bacteria were recovered overnight at 37°C and diluted in Brain Heart Infusion (BHI) broth (HuanKai Microbial, China) for further experiments. BHI was used as culture medium in planktonic bacteria assays. BHI supplemented with 1% (wt/vol) of sucrose (BHIS) was used as culture medium in biofilm assays. BHI supplemented with 1% (wt/vol) of agar (BHIA) was used in CFU counting assays. The negative control group was set as double distilled water of the same volume as BBR.

## Determination of minimum inhibitory concentration and minimum bactericidal concentration

The MIC and MBC of BBR against planktonic bacteria were determined by the reference protocol of the Clinical and Laboratory Standards Institute broth dilution method (57). Briefly, the overnight culture of *S. mutans* UA159, *S. mutans* ATCC 25175, *S. mutans* GS-5, *S. gordonii* DL-1, and *S. mitis* ATCC 6249 were diluted to $10^6$ CFU/mL in BHI broth and added to a 96-well round-bottom cell culture plate. BBR solution was then added to each well to achieve final concentrations ranging from 25 to 400 µg/mL. The plate was then incubated anaerobically at 37°C for 24 h. The MIC was defined as the lowest concentration of BBR that inhibited visible bacterial growth. At the termination of the MIC assay, a volume of the culture was struck on BHIA and incubated to observe growth. The MBC was defined as the lowest concentration that yielded no colony growth by subculturing on BHIA plates. Each experiment was performed with triplicate samples.

## Growth curve assay

Growth curve assay was conducted following a reported protocol (58) with modifications. Briefly, *S. mutans* UA159, *S. mutans* ATCC 25175, *S. mutans* GS-5, *S. gordonii* DL-1, and *S. mitis* ATCC 6249 were diluted to $10^6$ CFU/mL in BHI and treated with BBR solutions of varying final concentrations (200 µg/mL, 100 µg/mL, and 50 µg/mL) under anaerobic conditions at 37°C for 24 h. The optical density at 600 nm (OD600) of each well was recorded by a microplate reader (BioTek, USA) every hour throughout 24 h incubation period, and the bacterial growth curves were drawn using the recorded data. The assay was performed in triplicate.

## Crystal violet staining assay

In order to evaluate the impact of BBR on *S. mutans* UA159 biofilm formation, crystal violet staining assay was performed following reported protocols (59–61) with modifications. Briefly, *S. mutans* UA159 was diluted to $10^6$ CFU/mL in BHIS and treated with BBR solutions of varying final concentrations (200 µg/mL, 100 µg/mL, and 50 µg/mL) in a 24-well flat-bottom plate under anaerobic conditions at 37°C for 24 h. After incubation, the biofilms were fixed using anhydrous methanol for 15 min. The fixed biofilms were stained with 0.1% (wt/vol) crystal violet for 5 min. After removing the solution, an anhydrous ethanol solution was added to each well to dissolve the dye under gentle shaking for 30 min. Finally, the optical density at 575 nm (OD575) of each well was measured by a microplate reader (Epoch2, BioTek, USA). The assay was performed in triplicate.

## Quantitative assessment of water-insoluble EPS

The anthrone-sulfuric method was used to confirm the inhibitory effect of BBR on the synthesis of water-insoluble EPS by *S. mutans* following reported protocols (62, 63) with modifications. Briefly, *S. mutans* was incubated following the procedure outlined in Section Crystal violet staining assay. After incubation, the excess culture medium was removed, and the adherent *S. mutans* cells in biofilms were resuspended. The suspension was then centrifuged at 6,000 rpm, 4° for 5 min. The pellet was resuspended and washed three times with sterile phosphate-buffered solution (PBS; 0.1 M, pH 7.4) to remove all water-soluble EPS. The water-insoluble EPS was extracted using 0.4 M NaOH with constant agitation at 37°C for 15 min. The supernatant was then mixed with three volumes of anthrone-sulfuric acid reagent and heated in a water bath at 99°C until the reaction was completed. Afterward, the optical density at 625 nm (OD625) was measured using a microplate reader (Epoch2, BioTek, USA), and the concentrations of water insoluble EPS were calculated using a standard curve. The experiment was conducted in triplicate.

## Lactic acid assay

The acidogenicity of *S. mutans* under the effect of BBR was evaluated by lactic acid assay (64). Briefly, *S. mutans* was incubated following the procedure outlined in Section Crystal violet staining assay. After incubation, the excess culture medium was replaced with buffered peptone water (HuanKai Microbial, China) containing 0.2% sucrose for further incubation under anaerobic conditions at 37℃ for 2 h. The supernatants were tested using a lactic acid assay kit (Jiancheng, China) following the manufacturer's instructions. The optical density at 530 nm (OD530) was measured using a microplate reader (Epoch2, BioTek, USA), and lactate concentrations were calculated using a standard curve. The experiment was conducted in triplicate.

## Glycolytic pH drop assay

The impact of BBR on the pH drop of *S. mutans* glycolysis was measured by glycolytic pH drop assay. Briefly, *S. mutans* was harvested at mid-logarithmic phase via centrifugation, washed with a 50 mM KCl solution, and resuspended in the same salt solution. Glucose was added to each tube to achieve a final concentration of 0.1% and 1% (wt/vol). BBR was then added to each tube to achieve final concentrations of 200 µg/mL, 100 µg/mL, and 50 µg/mL. A suspension of *S. mutans* in salt solution was used as a negative control. The pH decrease resulting from the glycolytic activity of *S. mutans* was monitored at 30 min intervals over a period of 270 min using a pH meter (FE28, METTLER TOLEDO, Switzerland). The assay was performed in triplicate.

## Scanning electron microscopy

The effect of BBR on the surface morphology of *S. mutans* biofilm was observed by SEM. Briefly, *S. mutans* UA159 was diluted to $10^6$ CFU/mL in BHIS and treated with BBR solutions of varying final concentrations (200 µg/mL, 100 µg/mL, and 50 µg/mL) in a 24-well plate with a round-shaped glass coverslip at the bottom of each well under anaerobic conditions at 37℃ for 24 h. After removing the excess culture medium, the biofilms were fixed with 4% glutaraldehyde (EM Grade, Solarbio, China). The plate was then preserved at 4℃ for 12 h. After that, the biofilms were dehydrated using a series of ethanol concentrations (30%, 50%, 70%, 80%, 90%, 95%, and 100%, vol/vol). After gold sputtering, the samples were observed using a SEM (Zeiss Sigma 300, Germany). Three points were randomly selected for observation on each glass coverslip.

## Confocal laser scanning microscopy

CLSM was used to evaluate the quantitative correlation between live and dead bacteria, as well as live bacteria and EPS. Briefly, *S. mutans* UA159 was diluted to $10^6$ CFU/mL in BHIS and treated with BBR solutions of varying final concentrations (100 µg/mL and 50 µg/mL) in a 24-well plate with a round-shaped glass coverslip at the bottom of each well under anaerobic conditions at 37℃ for 24 h. For the relationship between live and dead bacteria, the L-7012 LIVE/DEAD BacLight Bacterial Viability Kit (Invitrogen, USA) containing SYTO 9 which dyed live cells with intact membranes green fluorescence and propidium iodide (PI) which dyed dead cells with damaged cell membrane red fluorescence was used for bacterial staining according to the manufacturer's instructions. The observation process was conducted by a CLSM (Olympus FV3000, Japan). Dual-channel scanning observations were conducted using a red channel (excitation wavelength: 561 nm, emission wavelength: 500–550 nm, HV: 550V, Gain: 1×, Offset: 3%) and a green channel (excitation wavelength: 488 nm, emission wavelength: 570–620 nm, HV: 570V, Gain: 1×, Offset: 3%). The statistical results of biofilm quantification were then analyzed by micro-image analysis using Image J COMSTAT software (NIH, USA). Three points on each coverslip were randomly selected for observation. What's more, *S. mutans* were cultured on glass coverslips under anaerobic conditions at 37℃ for 24 h to form mature biofilms. After that, BBR solution was added to each well to achieve final concentrations of 200 µg/mL, 100 µg/mL, and 50 µg/mL. The plate was then incubated under

anaerobic conditions at 37°C for an additional 30 min, followed by the same procedures as described above. When it came to the relationship between live bacteria and EPS, SYTO 9 and Concanavalin A-TRITC (Ruixibio, China) were used for bacteria labeling and EPS labeling, respectively. The procedures followed were consistent with those described above, with constant exciting laser intensity, background level, contrast, and electronic zoom in each experiment.

## CFU counting assay

The CFU counting assay was used to quantify the impact of BBR on biofilm formation and mature biofilm of *S. mutans* by counting the number of live bacteria in *S. mutans* biofilm as reported (62). For the biofilm formation assay, *S. mutans* was incubated following the procedure outlined in Section Crystal violet staining assay. After incubation, the adherent *S. mutans* cells in biofilms were resuspended and serially diluted from $10^4$-fold to $10^6$-fold. The suspensions were then plated onto BHIA plates and incubated under anaerobic conditions at 37°C for 48 h to determine CFU counts. When it came to the mature biofilm assay, overnight cultures of *S. mutans* were diluted in BHIS and incubated under anaerobic conditions at 37°C for 24 h to form mature biofilms. After that, BBR solution was added to each well to achieve final concentrations of 200 µg/mL, 100 µg/mL, and 50 µg/mL. The plate was then incubated under anaerobic conditions at 37°C for another 24 h followed by the same procedures as described above to determine CFU counts in each well. In order to evaluate the short-term effect of BBR, *S. mutans* UA159 was diluted to $10^6$ CFU/mL and treated with BBR solutions of varying final concentrations (200 µg/mL, 100 µg/mL, and 50 µg/mL) for 5 and 10 min. The suspensions were then diluted and plated onto BHIA plates and incubated under anaerobic conditions at 37°C for 24 h to determine CFU counts. The CFU counting procedures were performed in triplicate.

## Bacterial aggregation assay

The bacterial aggregation assay was conducted following a previously reported protocol (65) with modifications. Briefly, an overnight culture of *S. mutans* UA159 suspension was harvested by centrifugation at 6,000 rpm, 4°C for 5 min and resuspended to an optical density at 600 nm (OD600) of approximately 0.5, as determined by using a microplate reader (Epoch2, BioTek, USA). The initial OD600 was recorded. BBR was then added to each tube to achieve final concentrations of 200 µg/mL, 100 µg/mL, and 50 µg/mL. A suspension of *S. mutans* in sterile PBS was used as a negative control. Aggregation was determined by measuring the optical density at 600 nm (OD600) during 6 h of incubation at 37°C. The percentage of aggregation was calculated by the following equation:

$$\text{Aggregation rate} = \left(1 - \frac{\text{OD}_t}{\text{OD}_{\text{Initial}}}\right) \times 100\%$$

OD$_t$ represents the optical density at time $t$ = 1, 2, 3, 4, 5, or 6 h. The bacterial aggregation assay was performed in triplicate.

## Membrane potential assay

Membrane depolarization activity was measured using 3,3′-dipropylthiadicarbocyanine Iodide [DiSC3(5)] following a previously reported protocol (66). A suspension of *S. mutans* ($10^7$ CFU/mL) in PBS was incubated with 40 µM DiSC3(5) at 37°C for 1 h. BBR solution was then added to each well to achieve final concentrations of 200 µg/mL, 100 µg/mL, and 50 µg/mL, followed by incubation at 37°C for an additional 1.5 h. The fluorescence intensity was recorded using a microplate reader (Epoch2, BioTek, USA) at excitation and emission wavelengths of 622 nm and 670 nm. The experiment was repeated three times.

**TABLE 1** Sequences of primers used in this study

| Primers | Sequences |
| --- | --- |
| *S. mutans* 16S rRNA | Forward: 5′-CCATGTGTAGCGGTGAAATGC-3′ |
| | Reverse: 5′-TCATCGTTTACGGCGTGGAC-3′ |
| *GtfB* | Forward: 5′-AGCCGAAAGTTGGTATCGTCC-3′ |
| | Reverse: 5′-TGACGCTGTGTTTCTTGGCTC-3′ |
| *GtfC* | Forward: 5′-TTCCGTCCCTTATTGATGACATG-3′ |
| | Reverse: 5′-AATTGAAGCGGACTGGTTGCT-3′ |
| *GtfD* | Forward: 5′-TTGACGGTGTTCGTGTTGAT-3′ |
| | Reverse: 5′-AAAGCGATAGGCGCAGTTTA-3′ |
| *ldh* | Forward: 5′-AAAAACCAGGCGAAACTCGC-3′ |
| | Reverse: 5′-CTGAACGCGCATCAACATCA-3′ |
| *vicR* | Forward: 5′-CGTGTAAAAGCGCATCTTCG-3′ |
| | Reverse: 5′-AATGTTCACGCGTCATCACC-3′ |
| *liaR* | Forward: 5′-CATGAAGATTTAACAGCGCG-3′ |
| | Reverse: 5′-CGTCCTGTGGCACTAAATGA-3′ |
| *comD* | Forward: 5′-TTCCTGCAAACTCGATCATATAGG-3′ |
| | Reverse: 5′-TGCCAGTTCTGACTTGTTTAGGC-3′ |

## Transmission electron microscopy

The surface morphology of *S. mutans* planktonic cells under the effect of BBR was observed by TEM (67). Briefly, *S. mutans* UA159 was diluted in BHI and treated with BBR solutions of varying final concentrations (200 µg/mL, 100 µg/mL, and 50 µg/mL) under anaerobic conditions at 37°C for 30 min. After incubation, the bacterial cells were resuspended and centrifuged at 6,000 rpm, 4°C for 5 min. The pellet was washed three times with PBS and fixed with 4% glutaraldehyde at 4°C for 12 h. The bacterial cells were then collected by centrifugation at 15,000 rpm at 4°C for 20 min. The samples were treated with 3% glutaraldehyde, 1% osmium tetroxide, acetone, and epoxy and cut into thin sections using a microtome (Ultracut-E, Reichert-Jung, Austria). Finally, the thin sections were observed by TEM (HT7800, Hitachi, Japan).

## Quantitative real-time PCR

The impact of BBR on the expression levels of virulence genes (*gtfB*, *gtfC*, *gtfD*, *ldh*, *vicR*, *liaR*, and *comD*) in *S. mutans* was assessed by the qRT-PCR method. Briefly, *S. mutans* UA159 was harvested at mid-logarithmic phase via centrifugation, resuspended to an optical density at 600 nm (OD600) of approximately 0.2, and treated with BBR solutions of varying final concentrations (200 µg/mL, 100 µg/mL, and 50 µg/mL) under anaerobic conditions at 37°C for 2 h. After incubation, the bacterial cells were centrifuged at 6,000 rpm, 4°C for 5 min and digested with lysozyme (Solarbio, China) at 37°C for 30 min. The total RNA of *S. mutans* UA159 was then extracted using the RNA-Quick Purification Kit (ES science, China) following the manufacturer's instructions. RNA reverse transcription was performed using the PrimeScript RT Reagent Kit (TaKaRa, Japan). The relative mRNA expression levels of the genes were measured using qRT-PCR, with *S. mutans* UA159 16S rRNA as the internal control. The primers used in this study were synthesized by TsingKe Biotechnology, Inc. The primer sequences are listed in Table 1. The PCR procedures consisted of an initial denaturation at 95°C for 30 s, followed by 40 cycles of amplification consisting of denaturation at 95°C for 10 s, annealing at 60°C for 20 s, and extension at 72°C for 20 s. The qRT-PCR procedures were performed using LC480 (Roche, USA). The relative mRNA expression levels of the genes were determined using the $2^{-\Delta\Delta CT}$ method. The experiment was conducted in triplicate.

## Cytotoxicity effect on host cells

The biocompatibility of BBR on human gingival epithelial cells (HGECs), human oral keratinocytes (HOKs), human periodontal ligament cells (HPDLCs), human gingival

fibroblasts (HGFs), and macrophages (RAW 264.7 and THP-1) was evaluated by the Cell-Counting-Kit 8 (CCK-8) assay following the manufacturer's instructions. The HGEC (CP-H178), HOK (CP-H382), HPDLC (CP-H234), HGF (CP-H240), RAW 264.7 (CL-0190), and THP-1 (CL-0233) cell lines were provided by the Institute of Stomatological Research, Sun Yat-sen University. The cells were treated with BBR solution at final concentrations of 200 µg/mL, 100 µg/mL, and 50 µg/mL under 5% $CO_2$ at 37°C for 0.5 h and 8 h to evaluate the short- and long-term effects of BBR. After incubation, CCK-8 reagent (Cell-Counting-Kit 8, Gbcbio, China) was added to each well. The plate was then incubated at 37°C for 2 h, and the optical density at 450 nm (OD450) of each well was measured using a microplate reader (Epoch2, BioTek, USA). The CCK-8 assay was performed in triplicate.

## Statistical analysis

The data were analyzed using SPSS software (IBM SPSS Statistics 26, USA). The statistical results were presented as mean ± standard deviation (SD). After validating the equal variance assumptions of the data, one-way analysis of variance (ANOVA) was performed, followed by Dunnett's post hoc tests. A $P$ value $<$ 0.05 was considered statistically significant. The assays were performed in triplicate.

## Conclusions

In summary, this study reports on the inhibitory effect of BBR on the growth of planktonic *S. mutans*, as demonstrated by MIC, MBC, and growth curve assay. BBR was found to inhibit biofilm formation and exhibit antimicrobial effects on *S. mutans*, as demonstrated by crystal violet staining assay, SEM, CLSM, and CFU counting assay. The inhibitory effect of BBR on the toxicity of *S. mutans* was evaluated by the anthrone-sulfuric method, glycolytic pH drop assay, lactic acid assay, and qRT-PCR. This is the first study to elucidate the membrane-destructive bactericidal mechanism of BBR on *S. mutans* by bacterial aggregation assay, membrane potential assay and TEM. Moreover, the results of the Cell-Counting-Kit 8 assay showed that BBR exhibited favorable biocompatibility on human oral cells and macrophages. In conclusion, the study demonstrates that BBR, the main effective component of the Chinese herb *Coptis chinensis* Franch., has significant anticaries activity against *S. mutans* and has the potential to be further developed as a novel anticaries agent for clinical prevention and treatment of dental caries.

## ACKNOWLEDGMENTS

This research was funded by the National Natural Science Foundation of China (82170987, 82100995), the Natural Science Foundation of Guangdong Province (2021A1515012535), the Science and Technology Projects in Guangzhou (SL2023A04J01702).

## AUTHOR AFFILIATIONS

[1]Hospital of Stomatology, Guangdong Provincial Key Laboratory of Stomatology, Guanghua School of Stomatology, Sun Yat-sen University, Guangzhou, Guangdong, China
[2]Shandong Second Medical University, Weifang, Shandong, China

## AUTHOR ORCIDs

Chongmai Zeng http://orcid.org/0000-0003-0325-6469
Wentao Jiang http://orcid.org/0000-0002-1779-0438
Yang Cao http://orcid.org/0000-0003-1613-4516

## AUTHOR CONTRIBUTIONS

Chongmai Zeng, Conceptualization, Data curation, Formal analysis, Investigation, Methodology, Project administration, Software, Validation, Visualization, Writing –

original draft, Writing – review and editing | Wentao Jiang, Conceptualization, Data curation, Formal analysis, Funding acquisition, Investigation, Methodology, Project administration, Software, Validation, Visualization, Writing – original draft, Writing – review and editing | Chao Liu, Formal analysis, Validation, Writing – original draft | Rongcheng Yu, Formal analysis, Validation, Writing – original draft | Yang Li, Formal analysis, Validation, Writing – original draft | Peiru Li, Formal analysis, Validation, Writing – original draft.

## ADDITIONAL FILES

The following material is available online.

## Open Peer Review

**PEER REVIEW HISTORY (review-history.pdf).** An accounting of the reviewer comments and feedback.

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
