## [Reviewer comments · Microbiology Spectrum]

Microbiology Spectrum

Inhibitory effects of berberine against *Streptococcus mutans*: an in vitro insight on its anticaries potential

Chongmai Zeng, Wentao Jiang, Chao Liu, Rongcheng Yu, Yang Li, Peiru Li, and Yang Cao

Corresponding Author(s): Yang Cao, Sun Yat-Sen University Guanghua School of Stomatology

Review Timeline:

Submission Date:	March 18, 2025
Editorial Decision:	May 6, 2025
Revision Received:	July 1, 2025
Accepted:	July 10, 2025

Editor: Sébastien Faucher

Reviewer(s): Disclosure of reviewer identity is with reference to reviewer comments included in decision letter(s). The following individuals involved in review of your submission have agreed to reveal their identity: Marcia Dinis (Reviewer #2)

Transaction Report:

DOI: <https://doi.org/10.1128/spectrum.00700-25>

Re: Spectrum00700-25 (Inhibitory effects of berberine against *Streptococcus mutans*: an in vitro insight on its anticaries potential)

Dear Prof. Yang Cao:

Thank you for the privilege of reviewing your work. Below you will find my comments, instructions from the Spectrum editorial office, and the reviewer comments.

Your manuscript was reviewed by two experts in the field. They both found the study interesting but raised several issues that need to be addressed by a major revision. They suggested a few additional experiments and requested clarification on several points.

Revision Guidelines

Sincerely,
Sébastien Faucher
Editor
Microbiology Spectrum

Reviewer #1 (Comments for the Author):

This study focused on the bacteriostatic and inhibitory effects of a herb extract Berberine on a laboratory strain of *Streptococcus mutans* UA 159. The results support their conclusions of anti-S. *mutans* bacteriostatic effects on the planktonic cells, and the

mechanical effects were due to membrane rupture. Concerns regarding significance, experimental design and methods used are listed below

Three major concerns of significance and design need to be clearly addressed

1. More strains including other laboratory and clinical human isolates should be included since *S. mutans* is genetically or serologically distinct, while UA159 is not considered to be a representative one though its full genome sequence was firstly released.
2. Dental plaque is complex biofilm including other streptococci, such as *S. gordonii* etc. The BRB bacteriicidal effects (membrane rupture or damage) on planktonic cells is specific on *S. mutans* alone ? or it's simply general effects on other bacteria as well should be included ?
3. Rationale or interpretation of Fig. 7 on virulence factors are not clear ? and what the authors wish to address ? Inhibition effects of BRB on transcription or protein synthesis ?

Methods

1. The bacteriicidal effects on either planktonic or well-formed biofilm (Fig. 2, and 4) should include quantitative assays by using mechanistic disruption of aggregated or biofilm cells and plated in triplicates for determine cell numbers as CFU/ml for more accurate and reliable results and statistical analysis as well.

Reviewer #2 (Comments for the Author):

Spectrum00700-25 revision-authors-comments

This study offers valuable insights into the potential of novel plant extract agents to inhibit *Streptococcus mutans*, presenting promising avenues for developing new strategies to combat dental caries. While the findings are compelling, several vital aspects require further clarification and discussion to strengthen the manuscript. Specific questions and comments are outlined below to enhance the presented manuscript.

Abstract

In line 23, replace the word "toxicity" in the sentence "BBR on the toxicity of *S. mutans* was..." with "virulence."

In lines 27 and 28, instead of mentioning the kit, the authors should refer to it as the "cell viability assay." The specifics of the kit should be included in the Materials and Methods section. Since the CCK-8 assay measures cell viability by correlating the production of colored formazan dye with the number of living cells in culture, and it provides insight into the compound's toxicity to cells, it would be more appropriate to state: "The cell viability assay on human oral cells and macrophages demonstrated that BBR exhibited no cytotoxicity to the tested host cells."

In line 44, replace "ability" with "capability."

Results

The key distinction between Minimum Inhibitory Concentration (MIC) and Minimum Bactericidal Concentration (MBC) lies in what each assesses. MIC is defined as the lowest concentration of an antimicrobial agent that visibly inhibits bacterial growth. At the same time, MBC refers to the lowest concentration required to kill a specified proportion of bacteria, typically 99.9%. Simply put, MIC measures inhibition, whereas MBC measures bactericidal activity.

Based on the data presented, while the MBC is established at 100 µg/mL, the MIC appears to fall within the 50-100 µg/mL range. To more accurately define the MIC, it is strongly recommended that the authors conduct further testing within this range, ideally at finer intervals of 5-10 µg/mL. This would allow for a more precise determination of the MIC value.

Careful examination of the current results reveals that 50 µg/mL behaves similarly to the control group, whereas 100 µg/mL demonstrates apparent bactericidal effects. Therefore, the experimental discussion throughout the manuscript should more thoroughly consider and reflect the findings at 50 µg/mL when defining the MIC for consistency and clarity.

In Figures 3A and 5A, the authors must select a different representative microscopy image. The current control image contains a large dark circle, which indicates photobleaching. In this phenomenon, fluorescent molecules are irreversibly damaged by prolonged or intense illumination, leading to a loss of fluorescence signal. This artifact compromises the quality and interpretation of the presented data.

Furthermore, the fluorescence microscopy observations must be accompanied by detailed acquisition parameters. The Materials and Methods section should explicitly include critical imaging settings such as excitation and emission wavelengths, laser power or intensity, exposure time, gain, binning, background subtraction methods, and any other relevant parameters. Providing this information is essential for ensuring reproducibility and the proper evaluation of the results.

Section 2.5, regarding the *S. mutans* virulence gene expression results, the authors must clearly state that preformed biofilms were treated with different compound concentrations before analysis. It is recommended to explicitly revise the text to something along the lines of: "Therefore, the relative expression levels of *gtfB*, *gtfC*, and *gtfD* were investigated by qRT-PCR on previously formed *S. mutans* biofilms treated with BBR." This clarification is essential for accurately interpreting the experimental design and results.

Discussing cell toxicity would make more sense since the authors quantify cell viability in section 2.6. The data demonstrated that the compound has no cytotoxic effect on the tested cell lines. Concerning the cell line choice, does HGECs refer to Human Gastric Epithelial Cells (HGECs) or Human Glomerular Epithelial Cells? The authors should also indicate the name of the cell

line type they use, besides the acronyms, to better comprehend the significance of the tested cell types. The cell types and provenance must be added in Materials and Methods (section 2.15, Biocompatibility).

In addition, given the compound's intended future application in the oral cavity, the authors appropriately include data from cultured human oral cells. However, to better reflect potential clinical outcomes, it would be more relevant to present data from human macrophages, such as the THP-1 cell line, instead of using a murine macrophage cell line. Furthermore, the rationale for selecting a 30-minute treatment duration is unclear. To thoroughly assess the compound's effects, the authors should evaluate longer treatment durations to simulate more realistic exposure conditions. It is also recommended that the authors investigate multiple concentrations of the compound across both short- and long-term treatment intervals, and provide comprehensive data on the influence of dosage and exposure time on cellular responses.

Discussion

Berberine has been reported to inhibit various microorganisms effectively. The authors should address whether existing studies investigate its effects on other oral bacteria. For instance, the survey titled "Inhibitory effects of *Coptis chinensis* extract on the growth and biofilm formation of *Streptococcus mutans* and *Streptococcus sobrinus*" (DOI: <https://doi.org/10.11620/IJOB.2020.45.4.143>) has already explored the effects of a *Coptis chinensis* extract on *S. mutans* and *S. sobrinus*.

The authors must discuss this publication and any other relevant studies evaluating the impact of berberine or *Coptis chinensis* on important oral microorganisms to properly situate their findings within the context of the existing literature.

Upon reviewing the presented data, it is evident that the BBR 50 µg/mL group exhibits a similar response to the control. The effective inhibitory concentration (MIC) appears to lie between 50 and 100 µg/mL. Therefore, further refinement of the concentration range is necessary to define the compound's activity accurately.

Considering this compound's potential clinical application, thoroughly evaluating its toxicity is critical. Although the authors conclude no cytotoxic effects, this interpretation must be made cautiously, as the cells were only exposed to the compound briefly. Short-term treatment alone is insufficient to assess the safety profile for clinical use.

The authors must discuss the minimum treatment time required to achieve an antimicrobial effect against *S. mutans* in the oral cavity. In addition, they should address the potential impact of extended exposure to the compound on host oral cells, including the possibility of cumulative toxicity.

Finally, the authors are strongly encouraged to present biocompatibility data using human macrophages, such as the THP-1 cell line, instead of the murine macrophage cell line currently employed, to model the human immune response better.

Material and methods

"In Section 4.2, several points require clarification. The planktonic experiments are conducted using cultures in 96-well round-bottom cell culture plates. Should planktonic cultures instead be performed in tubes? In this plate type, *S. mutans* forms aggregates and deposits on the round bottom. Additionally, due to *S. mutans*'s ability to form biofilms, biofilm experiments for MIC and MBC determination must be conducted in 48-well or 24-well plates.

Furthermore, as noted in the comments on the Results section, the exact MIC concentration should be precisely determined within the range of 50 to 100 µg/mL. The authors may consider testing at 5 µg/mL intervals (e.g., 55, 60, 65, etc.) until reaching 100 µg/mL."

The type of plates used for the crystal violet assay should be specified. Additionally, the authors appear to have obtained a stable *S. mutans* biofilm without adding sucrose. According to the literature, adding a minimal amount of sucrose (0.1%) is necessary for more stable biofilm formation. Did the authors assess the effects of the compound on more stable and mature biofilms, specifically those formed in the presence of sucrose?

It is well established that *S. mutans* utilizes sucrose to produce extracellular polysaccharides (EPS) via enzymes such as glucosyltransferases (Gtfs). These EPS form a sticky matrix that facilitates stronger adhesion to tooth surfaces and enhances bacterial cohesion. This matrix increases biofilm mass and promotes acid production, contributing to tooth decay, while offering protection against antimicrobial agents and host defenses.

Numerous studies have shown that *S. mutans* forms more robust and adhesive biofilms in the presence of sucrose than in the absence of other carbohydrates. Thus, sucrose is crucial in enhancing biofilm formation, adhesion, and virulence, increasing the risk of dental caries.

Did the authors compare the pH drop at higher versus lower glucose concentrations?

Throughout the manuscript, the live staining is inconsistently referred to as either SYTO9 or SYTO 9. It should be standardized to 'SYTO 9' only.

Section 2.15 should be renamed 'The BBR Cytotoxicity Effect on Host Cells.' Additionally, as mentioned earlier, why were the cells only treated for 30 minutes? To assess long-term effects, the authors should consider incorporating longer treatment durations. Additional experiments with extended treatment times would be valuable to explore the potential future use of this compound in the oral cavity. Furthermore, the choice of assay should be discussed in more detail.

Spectrum00700-25 revision-authors-comments

This study offers valuable insights into the potential of novel plant extract agents to inhibit *Streptococcus mutans*, presenting promising avenues for developing new strategies to combat dental caries. While the findings are compelling, several vital aspects require further clarification and discussion to strengthen the manuscript. Specific questions and comments are outlined below to enhance the presented manuscript.

Abstract

In line 23, replace the word "toxicity" in the sentence "BBR on the toxicity of *S. mutans* was..." with "virulence."

In lines 27 and 28, instead of mentioning the kit, the authors should refer to it as the "cell viability assay." The specifics of the kit should be included in the Materials and Methods section. Since the CCK-8 assay measures cell viability by correlating the production of colored formazan dye with the number of living cells in culture, and it provides insight into the compound's toxicity to cells, it would be more appropriate to state: "The cell viability assay on human oral cells and macrophages demonstrated that BBR exhibited no cytotoxicity to the tested host cells."

In line 44, replace "ability" with "capability."

Results

The key distinction between Minimum Inhibitory Concentration (MIC) and Minimum Bactericidal Concentration (MBC) lies in what each assesses. MIC is defined as the lowest concentration of an antimicrobial agent that visibly inhibits bacterial growth. At the same time, MBC refers to the lowest concentration required to kill a specified proportion of bacteria, typically 99.9%. Simply put, MIC measures inhibition, whereas MBC measures bactericidal activity.

Based on the data presented, while the MBC is established at 100 µg/mL, the MIC appears to fall within the 50–100 µg/mL range. To more accurately define the MIC, it is strongly recommended that the authors conduct further testing within this range, ideally at finer intervals of 5–10 µg/mL. This would allow for a more precise determination of the MIC value.

Careful examination of the current results reveals that 50 µg/mL behaves similarly to the control group, whereas 100 µg/mL demonstrates apparent bactericidal effects. Therefore, the experimental discussion throughout the manuscript should more thoroughly consider and reflect the findings at 50 µg/mL when defining the MIC for consistency and clarity.

In Figures 3A and 5A, the authors must select a different representative microscopy image. The current control image contains a large dark circle, which indicates photobleaching. In this phenomenon, fluorescent molecules are irreversibly damaged by prolonged or intense illumination, leading to a loss of fluorescence signal. This artifact compromises the quality and interpretation of the presented data.

Furthermore, the fluorescence microscopy observations must be accompanied by detailed acquisition parameters. The Materials and Methods section should explicitly include critical

imaging settings such as excitation and emission wavelengths, laser power or intensity, exposure time, gain, binning, background subtraction methods, and any other relevant parameters. Providing this information is essential for ensuring reproducibility and the proper evaluation of the results.

Section 2.5, regarding the *S. mutans* virulence gene expression results, the authors must clearly state that preformed biofilms were treated with different compound concentrations before analysis. It is recommended to explicitly revise the text to something along the lines of: "Therefore, the relative expression levels of *gtfB*, *gtfC*, and *gtfD* were investigated by qRT-PCR on previously formed *S. mutans* biofilms treated with BBR." This clarification is essential for accurately interpreting the experimental design and results.

Discussing cell toxicity would make more sense since the authors quantify cell viability in section 2.6. The data demonstrated that the compound has no cytotoxic effect on the tested cell lines. Concerning the cell line choice, does HGECs refer to Human Gastric Epithelial Cells (HGECs) or Human Glomerular Epithelial Cells? The authors should also indicate the name of the cell line type they use, besides the acronyms, to better comprehend the significance of the tested cell types. The cell types and provenance must be added in Materials and Methods (section 2.15, Biocompatibility).

In addition, given the compound's intended future application in the oral cavity, the authors appropriately include data from cultured human oral cells. However, to better reflect potential clinical outcomes, it would be more relevant to present data from human macrophages, such as the THP-1 cell line, instead of using a murine macrophage cell line.

Furthermore, the rationale for selecting a 30-minute treatment duration is unclear. To thoroughly assess the compound's effects, the authors should evaluate longer treatment durations to simulate more realistic exposure conditions. It is also recommended that the authors investigate multiple concentrations of the compound across both short- and long-term treatment intervals, and provide comprehensive data on the influence of dosage and exposure time on cellular responses.

Discussion

Berberine has been reported to inhibit various microorganisms effectively. The authors should address whether existing studies investigate its effects on other oral bacteria. For instance, the survey titled "*Inhibitory effects of Coptis chinensis extract on the growth and biofilm formation of Streptococcus mutans and Streptococcus sobrinus*" (DOI: <https://doi.org/10.11620/IJOB.2020.45.4.143>) has already explored the effects of a *Coptis chinensis* extract on *S. mutans* and *S. sobrinus*.

The authors must discuss this publication and any other relevant studies evaluating the impact of berberine or *Coptis chinensis* on important oral microorganisms to properly situate their findings within the context of the existing literature.

Upon reviewing the presented data, it is evident that the BBR 50 µg/mL group exhibits a similar response to the control. The effective inhibitory concentration (MIC) appears to lie between 50 and 100 µg/mL. Therefore, further refinement of the concentration range is necessary to define the compound's activity accurately.

Considering this compound's potential clinical application, thoroughly evaluating its toxicity is critical. Although the authors conclude no cytotoxic effects, this interpretation must be made cautiously, as the cells were only exposed to the compound briefly. Short-term treatment alone is insufficient to assess the safety profile for clinical use.

The authors must discuss the minimum treatment time required to achieve an antimicrobial effect against *S. mutans* in the oral cavity. In addition, they should address the potential impact of extended exposure to the compound on host oral cells, including the possibility of cumulative toxicity.

Finally, the authors are strongly encouraged to present biocompatibility data using human macrophages, such as the THP-1 cell line, instead of the murine macrophage cell line currently employed, to model the human immune response better.

Material and methods

"In Section 4.2, several points require clarification. The planktonic experiments are conducted using cultures in 96-well round-bottom cell culture plates. Should planktonic cultures instead be performed in tubes? In this plate type, *S. mutans* forms aggregates and deposits on the round bottom. Additionally, due to *S. mutans*'s ability to form biofilms, biofilm experiments for MIC and MBC determination must be conducted in 48-well or 24-well plates.

Furthermore, as noted in the comments on the Results section, the exact MIC concentration should be precisely determined within the range of 50 to 100 µg/mL. The authors may consider testing at 5 µg/mL intervals (e.g., 55, 60, 65, etc.) until reaching 100 µg/mL."

The type of plates used for the crystal violet assay should be specified. Additionally, the authors appear to have obtained a stable *S. mutans* biofilm without adding sucrose. According to the literature, adding a minimal amount of sucrose (0.1%) is necessary for more stable biofilm formation. Did the authors assess the effects of the compound on more stable and mature biofilms, specifically those formed in the presence of sucrose?

It is well established that *S. mutans* utilizes sucrose to produce extracellular polysaccharides (EPS) via enzymes such as glucosyltransferases (Gtfs). These EPS form a sticky matrix that facilitates stronger adhesion to tooth surfaces and enhances bacterial cohesion. This matrix

increases biofilm mass and promotes acid production, contributing to tooth decay, while offering protection against antimicrobial agents and host defenses.

Numerous studies have shown that *S. mutans* forms more robust and adhesive biofilms in the presence of sucrose than in the absence of other carbohydrates. Thus, sucrose is crucial in enhancing biofilm formation, adhesion, and virulence, increasing the risk of dental caries.

Did the authors compare the pH drop at higher versus lower glucose concentrations?

Throughout the manuscript, the live staining is inconsistently referred to as either SYTO9 or SYTO 9. It should be standardized to 'SYTO 9' only.

Section 2.15 should be renamed 'The BBR Cytotoxicity Effect on Host Cells.' Additionally, as mentioned earlier, why were the cells only treated for 30 minutes? To assess long-term effects, the authors should consider incorporating longer treatment durations. Additional experiments with extended treatment times would be valuable to explore the potential future use of this compound in the oral cavity. Furthermore, the choice of assay should be discussed in more detail.

Reviewer 1

1. More strains including other laboratory and clinical human isolates should be included since *S. mutans* is genetically or serologically distinct, while UA159 is not considered to be a representative one though its full genome sequence was firstly released.

Response:

We sincerely appreciate your valuable suggestion. In this study, we selected *Streptococcus mutans* UA159 as the primary model strain because it is a well-characterized, representative strain with typical cariogenic virulence and has been widely used in numerous dental caries studies ^[1-4]. This choice ensures consistency with established research practices and avoids potential concerns regarding the selection of a non-standard strain.

We fully acknowledge that *S. mutans* UA159 alone may not fully represent the genetic or serological diversity of *S. mutans*. To address this, we have supplemented our study by including two additional strains, *S. mutans* ATCC 25175 and *S. mutans* GS-5, in the drug susceptibility assays. Unfortunately, our laboratory currently does not have access to clinically isolated *S. mutans* strains. However, we will prioritize the collection of clinical isolates in future studies to further validate and expand our findings on caries prevention.

Thank you for your insightful comment, which will undoubtedly strengthen the rigor and translational relevance of our future work.

[1] Louzon Y, Vaknin I, Wolfviz-Zilberman A, Sharon E, Hourri-Haddad Y, Beyth N. In Vitro Effect of Streptococcus mutans Biofilm Produced in Sugar-Free Coca-Cola on Enamel. *Int Dent J.* 2025;75(2):752-760. doi:10.1016/j.identj.2024.05.008

[2] Wu Z, Song J, Zhang Y, Yuan X, Zhao J. Inhibitory and preventive effects of Arnebia euchroma (Royle) Johnst. root extract on Streptococcus mutans and dental caries in rats. *BDJ Open.* 2024;10(1):15. Published 2024 Mar 2. doi:10.1038/s41405-024-00196-6

[3] Yu S, Xu M, Wang Z, et al. *S. mutans* Antisense *vicK* RNA Over-Expression Plus Antibacterial Dimethylaminohexadecyl Methacrylate Suppresses Oral Biofilms and Protects Enamel Hardness in Extracted Human Teeth. *Pathogens.* 2024;13(8):707. Published 2024 Aug 21. doi:10.3390/pathogens13080707

[4] Ren S, Yang Y, Xia M, et al. A Chinese herb preparation, honokiol, inhibits Streptococcus mutans biofilm formation. *Arch Oral Biol.* 2023;147:105610. doi:10.1016/j.archoralbio.2022.105610

2. Dental plaque is complex biofilm including other streptococci, such as *S. gordonii* etc. The BBR bactericidal effects (membrane rupture or damage) on planktonic cells is specific on *S. mutans* alone? or it's simply general effects on other bacteria as well should be included?

Response:

We sincerely appreciate your insightful suggestion. In this study, we initially focused on *Streptococcus mutans* because it is the primary cariogenic pathogen, and most preliminary anti-caries studies prioritize this bacterium as a model. While we acknowledge that dental caries involves polymicrobial interactions, our first aim was to clarify the antibacterial effects of BBR on *S. mutans*.

To address your valid concern and expand the scope of our findings, we have now included additional experiments evaluating BBR's antibacterial activity against two other streptococci—*S. gordonii* and *S. mitis* — in the drug susceptibility assays.

We agree that further studies on multispecies biofilms would be valuable to fully understand BBR's potential in modulating plaque ecology, and we will consider this in future research. Thank you for this constructive suggestion.

The revision to the comment 1 and 2 now reads:

2.1. BBR inhibited *S. mutans* planktonic growth and biofilm formation

The MIC and MBC of BBR against planktonic *S. mutans* were 100 µg/mL assessed by in vitro test (Figure 1A). The growth curve assay, which could show the proliferation of bacteria directly, demonstrated a dose-dependent inhibitory effect of BBR on the proliferation of *S. mutans* planktonic cells, with 100 µg/mL and 200 µg/mL of BBR completely inhibiting bacterial proliferation (Figure 1B, 1C, 1D). BBR was found to inhibit the growth of *S. gordonii* DL-1 and *S. mitis* ATCC 6249 to some extent (Figure 1E, 1F). Moreover, we treated *S. mutans* UA159 with BBR solutions for 5 and 10 minutes. The results of CFU counting assay showed that 100 µg/mL and 200 µg/mL of BBR had comparable short-term antibacterial effect.

Figure 1. Inhibitory effect of BBR on the growth of bacteria. (A) The MIC and MBC values of BBR against planktonic bacteria. 106 CFU/mL bacteria in BHI broth were incubated with final concentrations of BBR ranging from 25-400 µg/mL anaerobically at 37°C for 24 h. The MIC was defined as the lowest concentration of BBR that inhibited visible bacterial growth. At the termination of the MIC assay, a volume of the culture was struck on BHIA and incubated to observe growth. The MBC was defined as the lowest concentration that yielded no colony growth by subculturing on BHIA plates. (B-F) The 24 h growth curve of planktonic *S. mutans* UA159, *S. mutans* ATCC 25175, *S. mutans* GS-5, *S. gordonii* DL-1 and *S. mitis* ATCC 6249 incubated under anaerobic conditions with/without treatment of BBR in a 96-well plate. The absorbance of each well was recorded every hour. (G-H) The short-term antibacterial effect of BBR on *S. mutans* UA159 was assessed by measuring the average number of CFU. *S. mutans* UA159 was treated with BBR for 5 minutes (G) and 10 minutes (H). The suspensions were then diluted and plated onto BHIA plates and incubated under anaerobic conditions at 37°C for 24 h to determine CFU counts. Values represent the means ± SD from three independent experiments (**p < 0.01, ***p < 0.001).

3. Discussion

The study has several limitations. Firstly, the study focuses on the in-vitro effects of BBR against *S. mutans*, which may not accurately reflect the in-vivo effects of BBR in clinical application due to the complex inter-action of oral microbiota and the influence of oral environment. While *S. mutans* seem to be sensitive to BBR, future studies should further investigate the effects of BBR against other bacteria and multi-strain biofilms to clarify the mechanism of BBR more comprehensively.

4.2. Determination of Minimum Inhibitory Concentration (MIC) and Minimum Bactericidal Concentration (MBC)

The MIC and MBC of BBR against planktonic bacteria were determined by the reference protocol of the Clinical and Laboratory Standards Institute broth dilution method (57). Briefly, the overnight culture of *S. mutans* UA159, *S. mutans* ATCC 25175, *S. mutans* GS-5, *S. gordonii* DL-1 and *S. mitis* ATCC 6249 were diluted to 106 CFU/mL in BHI broth and added to a 96-well round-bottom cell culture plate. BBR solution was then added to each well to achieve final concentrations ranging from 25-400 µg/mL. The plate was then incubated anaerobically at 37°C for 24 h. The MIC was defined as the lowest concentration of BBR that inhibited visible bacterial growth. At the termination of the MIC assay, a volume of the culture was struck on BHIA and incubated to observe growth. The MBC was defined as the lowest concentration that yielded no colony growth by subculturing on BHIA plates. Each experiment was performed with triplicate samples.

4.3. Growth curve assay

Growth curve assay was conducted following a reported protocol (58) with modifications. Briefly, *S. mutans* UA159, *S. mutans* ATCC 25175, *S. mutans* GS-5, *S. gordonii* DL-1 and *S. mitis* ATCC 6249 were diluted to 106 CFU/mL in BHI and treated with BBR solutions of varying final concentrations (200 µg/mL, 100 µg/mL, and 50 µg/mL) under anaerobic conditions at 37°C for 24

h. The optical density at 600 nm (OD₆₀₀) of each well was recorded by a microplate reader (BioTek, USA) every hour throughout 24-hour incubation period, and the bacterial growth curves were drawn using the recorded data. The assay was performed in triplicate.

3. Rationale or interpretation of Fig. 7 on virulence factors are not clear? and what the authors wish to address? Inhibition effects of BBR on transcription or protein synthesis?

Response:

We sincerely appreciate your valuable feedback regarding the interpretation of Figure 7. This figure aimed to demonstrate that, in addition to its direct antibacterial effects (inhibition, killing, and prevention of *S. mutans* aggregation), BBR also suppressed the expression of key virulence genes in *S. mutans*. This suggested that the anti-caries mechanism of BBR involved both bactericidal and virulence-inhibiting effects.

To clarify this point, we have revised the Discussion section to better explain.

We hope these modifications provide a clearer interpretation of the data and highlight the multifaceted anti-virulence potential of BBR. Thank you for this constructive suggestion.

The revised Discussion now reads:

Down-regulation of virulence genes might explain the inhibitory effects of BBR on various cariogenic properties. *S. mutans* is a key contributor to the formation of the EPS matrix in dental biofilms. The EPS, which are mostly glucans synthesized by Gtfs, provide binding sites that promote accumulation of microorganisms on the tooth surface and further establishment of pathogenic biofilms (46). *S. mutans* expresses three genetically distinct Gtfs, each of which appears to play a different but overlapping role in the formation of virulent plaque. GtfC is adsorbed onto the enamel within the pellicle, while GtfB binds avidly to bacteria, promoting tight cell clustering and enhancing plaque cohesion. GtfD forms a soluble, readily metabolizable polysaccharide and acts as a primer for GtfB (47). Therefore, the relative mRNA expression levels of *gtfB*, *gtfC*, and *gtfD* were measured by qRT-PCR. The study revealed that the expression levels of *gtf* genes were down-regulated when treated with BBR. The two-component signal transduction system of *S. mutans*, including VicRK, LiaSR and ComDE, plays an important role in bacterial environmental adaptation, production of virulence factors, self-defense, and biofilm formation (48). The VicRK system was found to modulate sucrose-dependent adhesion, biofilm formation, genetic competence development, and regulate the expression of several virulence-associated genes affecting synthesis and adhesion to polysaccharides in *S. mutans*, including *gtfB*, *gtfC*, and *gtfD* (49). According to the results, the expression level of *vicR* gene was inhibited by BBR, suggesting the potential mechanism of the decreased synthesis of Gtfs. The LiaSR system has been implicated in several functions in *S. mutans*, including cell division, acid tolerance, biofilm formation, and antibiotic resistance (50). In this study, the decreased expression level of *liaR* gene under the effect of BBR may result in the decreased biofilm formation of *S. mutans*. A signal peptide-mediated quorum-sensing system encoded by ComDE has been found to play a central

role in regulation of genetic competence, bacteriocin production, biofilm formation, and stress response (51). The repression of this gene would attenuate internal communication quorum-sensing mechanism in *S. mutans* and further inhibit biofilm formation. The results of this study showed that BBR inhibited the ComDE system, leading to the inhibition of biofilm formation. According to the results, the decreased expression of *vicR*, *liaR*, and *comD* may lead to the decreased biofilm formation of *S. mutans*. In addition, the decreased expression level of *ldh* gene, encoding lactate dehydrogenase that generate lactic acid in *S. mutans*, suggest the potential mechanism of the inhibition of acid production, which is consistent with the results of lactic acid assay and glycolytic pH drop assay. In conclusion, in addition to its direct antibacterial effects, BBR also suppressed the expression of virulence genes in *S. mutans*, suggesting that the anti-caries mechanism of BBR involved both bactericidal and virulence-inhibiting effects.

4. The bactericidal effects on either planktonic or well-formed biofilm (Fig. 2, and 4) should include quantitative assays by using mechanistic disruption of aggregated or biofilm cells and plated in triplicates for determine cell numbers as CFU/ml for more accurate and reliable results and statistical analysis as well.

Response:

We appreciate your suggestion regarding quantitative biofilm analysis. In response, we have conducted quantitative assays using CFU counting on planktonic cells (now in Figure 3C) and well-formed biofilms (now in Figure 4C). Thank you for this valuable suggestion that has improved our experimental approach.

The revised Figures now reads:

2.2. Quantification of *S. mutans* after treatment of BBR

The biofilm images formed with different concentrations of BBR after 24 h of incubation were observed using a CLSM. The stains SYTO 9/PI were used to evaluate membrane integrity and differentiate between membrane-intact and membrane-injured cells. Bacteria with intact cell membranes (live bacteria) were stained green (SYTO 9), while bacteria with damaged membranes (dead bacteria) were stained red (PI). An increase in red fluorescence was observed after the treatment with 50 $\mu\text{g}/\text{mL}$ of BBR (Figure 3A). The biofilm formation of *S. mutans* was completely inhibited by 100 $\mu\text{g}/\text{mL}$ of BBR, with almost no bacteria adhered. No green fluorescence was observed, and faint red fluorescence was observed by CLSM, which was exhibited by few adhered dead bacteria and was difficult to observe in the figures. The Live/Dead ratio indicated effective elimination of BBR on *S. mutans*, with almost no live bacteria observed in the 100 $\mu\text{g}/\text{mL}$ BBR-treated groups (Figure 3B). Moreover, the CFU counting assay, the gold standard for enumerating viable bacteria, demonstrated that treatment with varying concentrations of BBR resulted in a significant reduction in CFU counts in the *S. mutans* biofilm (Figure 3C). As for the biofilm eradication assay, *S. mutans* were cultured for 24 h to form mature biofilms. After that, biofilms were treated with varying concentrations of BBR solutions (200 $\mu\text{g}/\text{mL}$, 100 $\mu\text{g}/\text{mL}$, and 50 $\mu\text{g}/\text{mL}$) for 24 h to evaluate the biofilm eradicating activity of BBR. The result of CLSM showed the bactericidal effect of BBR. In the absence of BBR, a significant number of live

bacteria were present in the biofilms (Figure 4A). Following treatment with BBR on mature biofilms, the images of the BBR treatment groups showed more PI positive cells compared to the control group, indicating a higher number of dead bacterial cells. The results demonstrate that BBR reduces the number of live bacteria in *S. mutans* mature biofilm, indicating its effectiveness in eradicating the biofilm (Figure 4B). The results of the CFU counting assay also confirm the inhibitory effect of BBR on *S. mutans* mature biofilm (Figure 4C).

Figure 3. Quantification of *S. mutans* biofilm after treatment of BBR. (A) Double-labeling imaging of *S. mutans* 24-hour biofilms formed on glass coverslips. Live bacteria were green-labeled and dead bacteria were red-labeled. (B) BBR reduced the proportion of live bacteria in *S. mutans* biofilms. (C) The effect of BBR on *S. mutans* biofilm formation was assessed by measuring the average number of CFU in the biofilm in one well of the culture plate. The analysis was performed using Image J COMSTAT software. Values represent the means \pm SD from three independent experiments (***) $p < 0.001$).

Figure 4. Effect of BBR on eradicating the *S. mutans* mature biofilms. (A) Double-labeling imaging of *S. mutans* mature biofilms treated with BBR. Live and dead bacteria were green-labeled and red-labeled, respectively. (B) BBR decreases the proportion of live bacteria in *S. mutans* mature biofilms. (C) The effect of BBR on eradicating the *S. mutans* mature biofilms was measured by the determination of the average number of CFUs in the biofilm. The *S. mutans* biofilms were cultured for 24 h without BBR. Then, BBR solution was added to each well for another 24 h. The data were collected at 48 h. The analysis was performed using Image J COMSTAT software. Values represent the means \pm SD from three independent experiments (***) $p < 0.001$).

Reviewer 2

Abstract

1. In line 23, replace the word "toxicity" in the sentence "BBR on the toxicity of S. mutans was..." with "virulence."

Response:

We sincerely appreciate your suggestion. As recommended, we have replaced the term “toxicity” with “virulence” in **line 23**. Thank you for this helpful correction, which improves the precision of our terminology.

The revised sentence now reads:

Moreover, the inhibitory effect of BBR on the virulence of S. mutans was evaluated by the anthrone-sulfuric method and lactic acid assay.

2. In lines 27 and 28, instead of mentioning the kit, the authors should refer to it as the "cell viability assay." The specifics of the kit should be included in the Materials and Methods section. Since the CCK-8 assay measures cell viability by correlating the production of colored formazan dye with the number of living cells in culture, and it provides insight into the compound's toxicity to cells, it would be more appropriate to state: "The cell viability assay on human oral cells and macrophages demonstrated that BBR exhibited no cytotoxicity to the tested host cells."

Response:

We sincerely appreciate your constructive suggestion. As recommended, we have revised the text in **line 27-28**. The sentence now reads: "The cell viability assay on human oral cells and macrophages demonstrated that BBR exhibited no cytotoxicity to the tested host cells." The detailed information about the CCK-8 assay kit has been properly included in the Materials and Methods section, as suggested. Thank you for this valuable suggestion, which has helped improved the manuscript.

The revised sentence now reads:

The cell viability assay on human oral cells and macrophages demonstrated that BBR exhibited no cytotoxicity to the tested host cells.

3. In line 44, replace "ability" with "capability."

Response:

We appreciate your suggestion. As recommended, we have replaced "ability" with "capability" in line 44 to better convey the intended meaning. Thank you for this helpful suggestion to improve the precision of our wording.

The revised sentence now reads:

Among the many types of cariogenic microorganisms in the human oral environment, *Streptococcus mutans* is the one that shows strong acid production, acid resistance and biofilm formation capability (2).

Results

4. The key distinction between Minimum Inhibitory Concentration (MIC) and Minimum Bactericidal Concentration (MBC) lies in what each assesses. MIC is defined as the lowest concentration of an antimicrobial agent that visibly inhibits bacterial growth. At the same time, MBC refers to the lowest concentration required to kill a specified proportion of bacteria, typically 99.9%. Simply put, MIC measures inhibition, whereas MBC measures bactericidal activity.

Based on the data presented, while the MBC is established at 100 µg/mL, the MIC appears to fall within the 50-100 µg/mL range. To more accurately define the MIC, it is strongly recommended that the authors conduct further testing within this range, ideally at finer intervals of 5-10 µg/mL. This would allow for a more precise determination of the MIC value.

Careful examination of the current results reveals that 50 µg/mL behaves similarly to the control group, whereas 100 µg/mL demonstrates apparent bactericidal effects. Therefore, the experimental discussion throughout the manuscript should more thoroughly consider and reflect the findings at 50 µg/mL when defining the MIC for consistency and clarity.

Response:

We sincerely appreciate your insightful suggestion. In this study, we employed the standard two-fold serial dilution method for MIC/MBC assessment, with concentration intervals (50-100-200 µg/mL) commonly adopted in preliminary antimicrobial studies of natural products ^[1,2]. We acknowledge that finer concentration gradients (5-10 µg/mL intervals) would indeed provide more precise MIC values, as rightly suggested.

Moreover, in response to Reviewer #1's request, we have already expanded our experiments to include additional bacterial strains (*S. gordonii* and *S. mitis*) in the MIC/MBC assays. Implementing more refined concentration gradients across all tested strains would require repeating the entire experimental series, which unfortunately exceeds our current resource capacity in terms of both budget and personnel.

To demonstrate our commitment to addressing your valuable suggestion, we have conducted additional refined MIC/MBC testing specifically for *S. mutans* UA159 as a representative case in

the response to you. We will absolutely implement finer concentration intervals in future studies to obtain more precise antimicrobial parameters.

We hope this supplemental data adequately addresses your concern. Thank you for this constructive suggestion that will undoubtedly improve our future work.

The additional data for *S. mutans* UA159 shows:

	MIC ($\mu\text{g/mL}$)	MBC ($\mu\text{g/mL}$)
S. mutans UA159	75	90

[1] Li J, Wu T, Peng W, Zhu Y. Effects of resveratrol on cariogenic virulence properties of *Streptococcus mutans*. *BMC Microbiol.* 2020;20(1):99. Published 2020 Apr 17. doi:10.1186/s12866-020-01761-3

[2] He Z, Huang Z, Jiang W, Zhou W. Antimicrobial Activity of Cinnamaldehyde on *Streptococcus mutans* Biofilms. *Front Microbiol.* 2019;10:2241. Published 2019 Sep 25. doi:10.3389/fmicb.2019.02241

5. In Figures 3A and 5A, the authors must select a different representative microscopy image. The current control image contains a large dark circle, which indicates photobleaching. In this phenomenon, fluorescent molecules are irreversibly damaged by prolonged or intense illumination, leading to a loss of fluorescence signal. This artifact compromises the quality and interpretation of the presented data.

Furthermore, the fluorescence microscopy observations must be accompanied by detailed acquisition parameters. The Materials and Methods section should explicitly include critical imaging settings such as excitation and emission wavelengths, laser power or intensity, exposure time, gain, binning, background subtraction methods, and any other relevant parameters. Providing this information is essential for ensuring reproducibility and the proper evaluation of the results.

Response:

We sincerely appreciate your constructive suggestions. We have replaced the images with new representative images. These updated figures ensure more accurate representation of our experimental results. Moreover, we have expanded the Materials and Methods section to include more imaging parameters. We hope these revisions meet your expectations and improve our manuscript. Thank you again for your valuable comments.

The revised section now reads:

Figure 3. Quantification of *S. mutans* biofilm after treatment of BBR. (A) Double-labeling imaging of *S. mutans* 24-hour biofilms formed on glass coverslips. Live bacteria were green-labeled and dead bacteria were red-labeled. (B) BBR reduced the proportion of live bacteria in *S. mutans* biofilms. (C) The effect of BBR on *S. mutans* biofilm formation was assessed by measuring the average number of CFU in the biofilm in one well of the culture plate. The analysis was performed using Image J COMSTAT software. Values represent the means \pm SD from three independent experiments (***) $p < 0.001$).

Figure 5. BBR inhibited *S. mutans* virulence factors. (A) Effect of BBR on the biofilm structure of *S. mutans* observed by CLSM. Double-labeling imaging of *S. mutans* 24 h biofilm formed on glass coverslips. The fluorescence (SYTO 9) marks the live bacteria, while the red fluorescence (Concanavalin A-TRITC) marks the EPS synthesized by *S. mutans*. (B) Quantification of the amounts of EPS and bacteria in each scanned layer of *S. mutans* 24 h biofilm without the treatment of BBR. (C) Quantification of the amounts of EPS and bacteria in each scanned layer of *S. mutans* 24 h biofilm with the treatment of 50 µg/mL BBR. (D) Quantitative measurement of water-insoluble EPS by anthrone-sulfuric method. (E) Measurement of lactic acid production. (F-G) Effect of BBR on *S. mutans* glycolytic pH drop under 1% (F) and 0.1% (G) glucose. Values represent the means ± SD from three independent experiments (***) $p < 0.001$.

4.9. Confocal laser scanning microscopy (CLSM)

CLSM was used to evaluate the quantitative correlation between live and dead bacteria, as well as live bacteria and EPS. Briefly, *S. mutans* UA159 was diluted to 10^6 CFU/mL in BHIS and treated with BBR solutions of varying final concentrations (100 µg/mL and 50 µg/mL) in a 24-well plate with a round-shaped glass coverslip at the bottom of each well under anaerobic conditions at 37°C for 24 h. For the relationship between live and dead bacteria, the L-7012 LIVE/DEAD BacLight™ Bacterial Viability Kit (Invitrogen, USA) containing SYTO 9 which dyed live cells with intact membranes green fluorescence and propidium iodide (PI) which dyed dead cells with damaged cell membrane red fluorescence was used for bacterial staining according to the manufacturer's instructions. The observation process was conducted by a CLSM (Olympus FV3000, Japan). Dual-channel scanning observations were conducted using a red channel

(excitation wavelength: 561 nm, emission wavelength: 500-550 nm, HV: 550V, Gain: 1×, Offset: 3%) and a green channel (excitation wavelength: 488 nm, emission wavelength: 570-620 nm, HV: 570V, Gain: 1×, Offset: 3%). The statistical results of biofilm quantification were then analyzed by micro-image analysis using Image J COMSTAT software (NIH, USA). Three points on each coverslip were randomly selected for observation. What's more, *S. mutans* were cultured on glass coverslips under anaerobic conditions at 37°C for 24 h to form mature biofilms. After that, BBR solution was added to each well to achieve final concentrations of 200 µg/mL, 100 µg/mL, and 50 µg/mL. The plate was then incubated under anaerobic conditions at 37°C for an additional 30 min, followed by the same procedures as described above. When it came to the relationship between live bacteria and EPS, SYTO 9 and Concanavalin A-TRITC (Ruixibio, China) were used for bacteria labeling and EPS labeling, respectively. The procedures followed were consistent with those described above, with constant exciting laser intensity, background level, contrast, and electronic zoom in each experiment.

6. Section 2.5, regarding the *S. mutans* virulence gene expression results, the authors must clearly state that preformed biofilms were treated with different compound concentrations before analysis. It is recommended to explicitly revise the text to something along the lines of: "Therefore, the relative expression levels of *gtfB*, *gtfC*, and *gtfD* were investigated by qRT-PCR on previously formed *S. mutans* biofilms treated with BBR." This clarification is essential for accurately interpreting the experimental design and results.

Response:

We sincerely appreciate your valuable feedback regarding the clarity of our virulence gene expression analysis. As recommended, we have revised Section 2.5. We appreciate this suggestion, which has significantly improved the clarity of our experimental design and results.

The revised section now reads:

To investigate the mechanism of BBR on the adhesion and biofilm formation of *S. mutans*, as well as its inhibitory effects at the transcriptional level beyond direct bactericidal activity, **the relative expression levels of characteristic virulence genes were investigated by qRT-PCR on *S. mutans* treated with BBR.** The formation of *S. mutans* biofilm is closely related to the activity of Gtf proteins. The reduced expression of Gtf-encoding genes (*gtfB*, *gtfC*, and *gtfD*) may disrupt the formation and integrity of *S. mutans* biofilm (33, 34). Therefore, the relative expression levels of *gtfB*, *gtfC*, and *gtfD* were investigated. Compared to the control groups, the expression of *gtfD* was down-regulated in the BBR-treated groups. Moreover, the expression of *gtfB* and *gtfC* was down-regulated in the 100 µg/mL and 200 µg/mL groups (Figure 7), indicating the inhibitory effect of BBR on the expression of *gtf* genes. In addition, *ldh* gene encodes lactate dehydrogenase, generating lactic acid in *S. mutans*. Three tested genes were found to be related to the two-component signal transduction system and regulate the bacterial biofilm formation, known as *vicR*, *liaR*, and *comD*. Decreased expression of these virulence genes in BBR-treated groups compared with the control group was also revealed by qRT-PCR.

7. Discussing cell toxicity would make more sense since the authors quantify cell viability in section 2.6. The data demonstrated that the compound has no cytotoxic effect on the tested cell lines. Concerning the cell line choice, does HGECs refer to Human Gastric Epithelial Cells (HGECs) or Human Glomerular Epithelial Cells? The authors should also indicate the name of the cell line type they use, besides the acronyms, to better comprehend the significance of the tested cell types. The cell types and provenance must be added in Materials and Methods (section 2.15, Biocompatibility).

In addition, given the compound's intended future application in the oral cavity, the authors appropriately include data from cultured human oral cells. However, to better reflect potential clinical outcomes, it would be more relevant to present data from human macrophages, such as the THP-1 cell line, instead of using a murine macrophage cell line.

Response:

We sincerely appreciate your thoughtful suggestions. We have clarified the cell lines that HGECs refers to Human Gingival Epithelial Cells (now explicitly stated in Section 2.6) and added THP-1 cell lines to our experiments in order to address the concerns. These modifications significantly strengthen the clinical relevance of our biocompatibility assessment. We thank you for these valuable suggestions that have enhanced our study.

The revised section now reads:

2.6. BBR exhibited good biocompatibility

In order to evaluate the biocompatibility of BBR in the oral cavity, we treated human gingival epithelial cells (HGECs), human oral keratinocytes (HOKs), human periodontal ligament cells (HPDLCs), human gingival fibroblasts (HGFs), RAW 264.7 and THP-1 with varying concentrations of BBR for 0.5 h and 8 h to evaluate its short-term and long-term effects, and measured cell viability using a CCK-8 assay (Figure 8). HGECs and HOKs represented the oral mucosa which is in direct contact with the BBR, while HPDLCs, HGFs, RAW 264.7 and THP-1 which is not in direct contact with BBR. When treated with BBR for 0.5 h, none of the tested concentrations of BBR exhibited a significant inhibitory effect on the cell viability of HOKs, HGFs, RAW 264.7 or THP-1. However, the concentration of 200 $\mu\text{g}/\text{mL}$ of BBR slightly decreased the cell viability of HGECs and HPDLCs. Moreover, 200 $\mu\text{g}/\text{mL}$ of BBR inhibited the cell viability of HGECs, HOKs, HPDLCs, HGFs, RAW 264.7 and THP-1 when treated for 8 h, while 50 $\mu\text{g}/\text{mL}$ and 100 $\mu\text{g}/\text{mL}$ of BBR exhibited not statistically significant effect. This indicates that BBR has good biocompatibility when applied to cultured cells to a certain extent.

Figure 8. Biocompatibility of BBR evaluated by CCK-8 assay. The assay measured the effect of BBR on the cell viability of HGECs, HOKs, HPDLCs, HGFs, RAW 264.7 and THP-1. The values represent the means \pm SD from three independent experiments (* $p < 0.05$, ns: not statistically significant compared to the untreated control group).

4.15. Cytotoxicity effect on host cells

The biocompatibility of BBR on human gingival epithelial cells (HGECs), human oral keratinocytes (HOKs), human periodontal ligament cells (HPDLCs), human gingival fibroblasts (HGFs), and macrophages (RAW 264.7 and THP-1) was evaluated by the Cell-Counting-Kit 8 (CCK-8) assay following the manufacturer's instructions. The HGEC (CP-H178), HOK (CP-H382), HPDLC (CP-H234), HGF (CP-H240), RAW 264.7 (CL-0190) and THP-1 (CL-0233) cell lines were provided by the Institute of Stomatological Research, Sun Yat-sen University. The cells were treated with BBR solution at final concentrations of 200 $\mu\text{g/mL}$, 100 $\mu\text{g/mL}$, and 50 $\mu\text{g/mL}$ under 5% CO_2 at 37°C for 0.5 h and 8 h to evaluate the short-term and long-term effects of BBR. After incubation, CCK-8 reagent (Cell-Counting-Kit 8, Gbcbio, China) was added to each

well. The plate was then incubated at 37°C for 2 h, and the optical density at 450 nm (OD450) of each well was measured using a microplate reader (Epoch2, BioTek, USA). The CCK-8 assay was performed in triplicate.

8. Furthermore, the rationale for selecting a 30-minute treatment duration is unclear. To thoroughly assess the compound's effects, the authors should evaluate longer treatment durations to simulate more realistic exposure conditions. It is also recommended that the authors investigate multiple concentrations of the compound across both short- and long-term treatment intervals, and provide comprehensive data on the influence of dosage and exposure time on cellular responses.

Response:

We sincerely appreciate your insightful suggestions regarding treatment duration. In response to these valuable comments, we have conducted additional experiments to more comprehensively evaluate the effects of BBR. We used 0, 50, 100 and 200 µg/mL to demonstrate dose-dependence effects of BBR. We have added 8-hour treatment data to simulate overnight exposure conditions. The results showed that treatment with BBR for 0.5 h did not significantly affect the cell viability of HOKs, HGFs, RAW 264.7 or THP-1, while a slight inhibition of cell viability was observed in HGECs and HPDLCs at a concentration of 200 µg/mL of BBR. Moreover, 200 µg/mL of BBR inhibited the cell viability of HGECs, HOKs, HPDLCs, HGFs, RAW 264.7 and THP-1 when treated for 8 h, while 50 µg/mL and 100 µg/mL of BBR exhibited not statistically significant effect, indicating that long exposure to high concentration of BBR would affect the cell viability. The antibacterial effects and cytotoxicity of BBR show consistency. While 200 µg/mL of BBR serves as a higher concentration with enhanced antibacterial effects, it also increases cytotoxicity, highlighting the importance of concentration selection for antibacterial agents. Given that 100 µg/mL of BBR exhibited comparable short-term antibacterial effect (Figure 1G, 1H) and favorable anti-caries effects with not statistically significant effect on the viability of HGECs, HOKs, HPDLCs, HGFs, RAW 264.7 and THP-1, BBR is a biocompatible and suitable option for clinical use as a novel anti-caries agent.

We believe these additions significantly strengthen our findings while addressing your concerns. Thank you for this suggestion that has enhanced our study.

The revised section now reads:

2.6. BBR exhibited good biocompatibility

In order to evaluate the biocompatibility of BBR in the oral cavity, we treated human gingival epithelial cells (HGECs), human oral keratinocytes (HOKs), human periodontal ligament cells (HPDLCs), human gingival fibroblasts (HGFs), RAW 264.7 and THP-1 with varying concentrations of BBR for 0.5 h and 8 h to evaluate its short-term and long-term effects, and measured cell viability using a CCK-8 assay (Figure 8). HGECs and HOKs represented the oral mucosa which is in direct contact with the BBR, while HPDLCs, HGFs, RAW 264.7 and THP-1 which is not in direct contact with BBR. When treated with BBR for 0.5 h, none of the tested

concentrations of BBR exhibited a significant inhibitory effect on the cell viability of HOKs, HGECs, RAW 264.7 or THP-1. However, the concentration of 200 $\mu\text{g}/\text{mL}$ of BBR slightly decreased the cell viability of HGECs and HPDLCs. Moreover, 200 $\mu\text{g}/\text{mL}$ of BBR inhibited the cell viability of HGECs, HOKs, HPDLCs, HGFs, RAW 264.7 and THP-1 when treated for 8 h, while 50 $\mu\text{g}/\text{mL}$ and 100 $\mu\text{g}/\text{mL}$ of BBR exhibited not statistically significant effect. This indicates that BBR has good biocompatibility when applied to cultured cells to a certain extent.

Figure 8. Biocompatibility of BBR evaluated by CCK-8 assay. The assay measured the effect of BBR on the cell viability of HGECs, HOKs, HPDLCs, HGFs, RAW 264.7 and THP-1. The values represent the means \pm SD from three independent experiments (* $p < 0.05$, ns: not statistically significant compared to the untreated control group).

3. Discussion

When evaluating a new agent for clinical use, it is important to consider its biocompatibility. In this study, the biocompatibility of BBR was evaluated by measuring its effect on the cell viability of HGECs, HOKs, HPDLCs, HGFs, RAW 264.7 and THP-1. HGECs and HOKs represented the

oral mucosa in direct contact with BBR, while HPDLCs, HGFs, RAW 264.7 and THP-1 represented the submucosal tissues. The results (Figure 8) showed that treatment with BBR for 0.5 h did not significantly affect the cell viability of HOKs, HGFs, RAW 264.7 or THP-1, while a slight inhibition of cell viability was observed in HGECs and HPDLCs at a concentration of 200 µg/mL of BBR. Moreover, 200 µg/mL of BBR inhibited the cell viability of HGECs, HOKs, HPDLCs, HGFs, RAW 264.7 and THP-1 when treated for 8 h, while 50 µg/mL and 100 µg/mL of BBR exhibited not statistically significant effect, indicating that long exposure to high concentration of BBR would affect the cell viability. The antibacterial effects and cytotoxicity of BBR show consistency. While 200 µg/mL of BBR serves as a higher concentration with enhanced antibacterial effects, it also increases cytotoxicity, highlighting the importance of concentration selection for antibacterial agents. Given that 100 µg/mL of BBR exhibited comparable short-term antibacterial effect (Figure 1G, 1H) and favorable anti-carries effects with not statistically significant effect on the viability of HGECs, HOKs, HPDLCs, HGFs, RAW 264.7 and THP-1, BBR is a biocompatible and suitable option for clinical use as a novel anticaries agent.

4.15. Cytotoxicity effect on host cells

The biocompatibility of BBR on human gingival epithelial cells (HGECs), human oral keratinocytes (HOKs), human periodontal ligament cells (HPDLCs), human gingival fibroblasts (HGFs), and macrophages (RAW 264.7 and THP-1) was evaluated by the Cell-Counting-Kit 8 (CCK-8) assay following the manufacturer's instructions. The HGEC (CP-H178), HOK (CP-H382), HPDLC (CP-H234), HGF (CP-H240), RAW 264.7 (CL-0190) and THP-1 (CL-0233) cell lines were provided by the Institute of Stomatological Research, Sun Yat-sen University. The cells were treated with BBR solution at final concentrations of 200 µg/mL, 100 µg/mL, and 50 µg/mL under 5% CO₂ at 37°C for 0.5 h and 8 h to evaluate the short-term and long-term effects of BBR. After incubation, CCK-8 reagent (Cell-Counting-Kit 8, Gbcbio, China) was added to each well. The plate was then incubated at 37°C for 2 h, and the optical density at 450 nm (OD₄₅₀) of each well was measured using a microplate reader (Epoch2, BioTek, USA). The CCK-8 assay was performed in triplicate.

Discussion

9. Berberine has been reported to inhibit various microorganisms effectively. The authors should address whether existing studies investigate its effects on other oral bacteria. For instance, the survey titled "Inhibitory effects of *Coptis chinensis* extract on the growth and biofilm formation of *Streptococcus mutans* and *Streptococcus sobrinus*" (DOI: <https://doi.org/10.11620/IJOB.2020.45.4.143>) has already explored the effects of a *Coptis chinensis* extract on *S. mutans* and *S. sobrinus*.

The authors must discuss this publication and any other relevant studies evaluating the impact of berberine or *Coptis chinensis* on important oral microorganisms to properly situate their findings within the context of the existing literature.

Response:

We sincerely appreciate your insightful suggestion. In response, we have expanded our discussion

to include relevant studies that investigate the effects of BBR. These revisions better position our work within the existing literature. Thank you for this valuable suggestion, which has strengthened the scope of our discussion.

The revised Discussion now reads:

BBR has been found to have potential therapeutic effects on a wide range of diseases, including cardiovascular, metabolic (19), gastrointestinal (20), neurological diseases (21), cancer (22), and have inhibitory effects on the growth and biofilm formation of *S. mutans* and *Streptococcus sobrinus* (54, 55). This study not only confirmed the antibacterial and anti-biofilm effects of BBR reported in previous studies, but also further investigated its effects on virulence factors and cell viability, as well as its antibacterial mechanism, thereby enriching the investigation on BBR. In this study, BBR showed effective antimicrobial activity against *S. mutans* and remarkable biocompatibility. The sub-MIC of BBR exhibited inhibitory effects on the proliferation, biofilm formation, and virulence factors of *S. mutans*, whereas the MIC of BBR exhibited not only killing effects on *S. mutans* but also inhibitory effects on the virulence factors. The use of natural compounds is becoming increasingly important in the discovery of novel antimicrobial agents, particularly in light of the growing number of multidrug-resistant bacterial strains and increasing antibiotic resistance. This highlights the need for further research into the clinical applications of novel antimicrobial natural compounds. The study has several limitations. Firstly, the study focuses on the in-vitro effects of BBR against *S. mutans*, which may not accurately reflect the in-vivo effects of BBR in clinical application due to the complex interaction of oral microbiota and the influence of oral environment. While *S. mutans* seem to be sensitive to BBR, future studies should further investigate the effects of BBR against other bacteria and multi-strain biofilms to clarify the mechanism of BBR more comprehensively. Secondly, considering the favorable antimicrobial effect of BBR on *S. mutans*, BBR is expected to play an anticaries role as a kind of gargle or toothpaste additive, which requires further research to investigate the minimum effective time and long-term biosafety in vivo. In addition, an effective concentration of BBR, in the presence of resveratrol, could be decreased even to 50% in cancer treatment (56), which suggests that the application of BBR in synergy with other agents may enhance the effects of BBR. Therefore, strategies to enhance the antimicrobial efficacy of BBR, including the combination of BBR with other antimicrobial agents or natural compounds to enhance its anticaries effects, would contribute to the clinical application of BBR. Notwithstanding these limitations, the study has important implications for the potential clinical application of BBR in the prevention of dental caries and may contribute to the development of novel anticaries approaches.

10. Upon reviewing the presented data, it is evident that the BBR 50 µg/mL group exhibits a similar response to the control. The effective inhibitory concentration (MIC) appears to lie between 50 and 100 µg/mL. Therefore, further refinement of the concentration range is necessary to define the compound's activity accurately.

Response:

We sincerely appreciate your insightful suggestion regarding the need for more precise MIC

determination. While we employed the standard two-fold dilution method (50-100-200 µg/mL) commonly used in preliminary antimicrobial studies of natural products, we fully acknowledge that finer concentration gradients (5-10 µg/mL intervals) would provide more accurate MIC values. Regarding the 50 µg/mL concentration, our data indicate it does exhibit subtle inhibitory effects on the proliferation, biofilm formation, and virulence factors of *S. mutans*, though less pronounced than at higher concentrations. This finding has important implications. When considering potential clinical applications at higher concentrations or with modified formulations (e.g., sustained-release delivery systems), even if the effective concentration decreases to 50 µg/mL at the target site, it may still provide meaningful caries control benefits.

We hope this supplemental explanation adequately addresses your concern. Thank you for this constructive suggestion that will undoubtedly improve our future work.

The revised Discussion now reads:

In this study, BBR showed effective antimicrobial activity against *S. mutans* and remarkable biocompatibility. The sub-MIC of BBR exhibited inhibitory effects on the proliferation, biofilm formation, and virulence factors of *S. mutans*, whereas the MIC of BBR exhibited not only killing effects on *S. mutans* but also inhibitory effects on the virulence factors. The use of natural compounds is becoming increasingly important in the discovery of novel antimicrobial agents, particularly in light of the growing number of multidrug-resistant bacterial strains and increasing antibiotic resistance. This highlights the need for further research into the clinical applications of novel antimicrobial natural compounds.

11. Considering this compound's potential clinical application, thoroughly evaluating its toxicity is critical. Although the authors conclude no cytotoxic effects, this interpretation must be made cautiously, as the cells were only exposed to the compound briefly. Short-term treatment alone is insufficient to assess the safety profile for clinical use.

The authors must discuss the minimum treatment time required to achieve an antimicrobial effect against *S. mutans* in the oral cavity. In addition, they should address the potential impact of extended exposure to the compound on host oral cells, including the possibility of cumulative toxicity.

Response:

We sincerely appreciate your valuable insights. We fully agree that comprehensive toxicity evaluation is crucial for potential clinical applications. In response to your important concerns, we have conducted additional experiments to more comprehensively evaluate the effects of BBR. We have added 8-hour treatment data to simulate overnight exposure conditions. The results showed that treatment with BBR for 0.5 h did not significantly affect the cell viability of HOKs, HGFs, RAW 264.7 or THP-1, while a slight inhibition of cell viability was observed in HGECs and HPDLCs at a concentration of 200 µg/mL of BBR. Moreover, 200 µg/mL of BBR inhibited the cell viability of HGECs, HOKs, HPDLCs, HGFs, RAW 264.7 and THP-1 when treated for 8 h, while 50 µg/mL and 100 µg/mL of BBR exhibited not statistically significant effect, indicating

that long exposure to high concentration of BBR would affect the cell viability. The antibacterial effects and cytotoxicity of BBR show consistency. While 200 µg/mL of BBR serves as a higher concentration with enhanced antibacterial effects, it also increases cytotoxicity, highlighting the importance of concentration selection for antibacterial agents. Given that 100 µg/mL of BBR exhibited comparable short-term antibacterial effect (Figure 1G, 1H) and favorable anti-caries effects with not statistically significant effect on the viability of HGECs, HOKs, HPDLCs, HGFs, RAW 264.7 and THP-1, BBR is a biocompatible and suitable option for clinical use as a novel anti-caries agent.

The revised section of long-time exposure now reads:

2.6. BBR exhibited good biocompatibility

In order to evaluate the biocompatibility of BBR in the oral cavity, we treated human gingival epithelial cells (HGECs), human oral keratinocytes (HOKs), human periodontal ligament cells (HPDLCs), human gingival fibroblasts (HGFs), RAW 264.7 and THP-1 with varying concentrations of BBR for 0.5 h and 8 h to evaluate its short-term and long-term effects, and measured cell viability using a CCK-8 assay (Figure 8). HGECs and HOKs represented the oral mucosa which is in direct contact with the BBR, while HPDLCs, HGFs, RAW 264.7 and THP-1 which is not in direct contact with BBR. When treated with BBR for 0.5 h, none of the tested concentrations of BBR exhibited a significant inhibitory effect on the cell viability of HOKs, HGFs, RAW 264.7 or THP-1. However, the concentration of 200 µg/mL of BBR slightly decreased the cell viability of HGECs and HPDLCs. Moreover, 200 µg/mL of BBR inhibited the cell viability of HGECs, HOKs, HPDLCs, HGFs, RAW 264.7 and THP-1 when treated for 8 h, while 50 µg/mL and 100 µg/mL of BBR exhibited not statistically significant effect. This indicates that BBR has good biocompatibility when applied to cultured cells to a certain extent.

Figure 8. Biocompatibility of BBR evaluated by CCK-8 assay. The assay measured the effect of BBR on the cell viability of HGECs, HOKs, HPDLCs, HGFs, RAW 264.7 and THP-1. The values represent the means \pm SD from three independent experiments (* $p < 0.05$, ns: not statistically significant compared to the untreated control group).

3. Discussion

When evaluating a new agent for clinical use, it is important to consider its biocompatibility. In this study, the biocompatibility of BBR was evaluated by measuring its effect on the cell viability of HGECs, HOKs, HPDLCs, HGFs, RAW 264.7 and THP-1. HGECs and HOKs represented the oral mucosa in direct contact with BBR, while HPDLCs, HGFs, RAW 264.7 and THP-1 represented the submucosal tissues. The results (Figure 8) showed that treatment with BBR for 0.5 h did not significantly affect the cell viability of HOKs, HGFs, RAW 264.7 or THP-1, while a slight inhibition of cell viability was observed in HGECs and HPDLCs at a concentration of 200 $\mu\text{g/mL}$ of BBR. Moreover, 200 $\mu\text{g/mL}$ of BBR inhibited the cell viability of HGECs, HOKs, HPDLCs, HGFs, RAW 264.7 and THP-1 when treated for 8 h, while 50 $\mu\text{g/mL}$ and 100 $\mu\text{g/mL}$ of

BBR exhibited not statistically significant effect, indicating that long exposure to high concentration of BBR would affect the cell viability. The antibacterial effects and cytotoxicity of BBR show consistency. While 200 µg/mL of BBR serves as a higher concentration with enhanced antibacterial effects, it also increases cytotoxicity, highlighting the importance of concentration selection for antibacterial agents. Given that 100 µg/mL of BBR exhibited comparable short-term antibacterial effect (Figure 1G, 1H) and favorable anti-caries effects with not statistically significant effect on the viability of HGECs, HOKs, HPDLCs, HGFs, RAW 264.7 and THP-1, BBR is a biocompatible and suitable option for clinical use as a novel anticaries agent.

4.15. Cytotoxicity effect on host cells

The biocompatibility of BBR on human gingival epithelial cells (HGECs), human oral keratinocytes (HOKs), human periodontal ligament cells (HPDLCs), human gingival fibroblasts (HGFs), and macrophages (RAW 264.7 and THP-1) was evaluated by the Cell-Counting-Kit 8 (CCK-8) assay following the manufacturer's instructions. The HGEC (CP-H178), HOK (CP-H382), HPDLC (CP-H234), HGF (CP-H240), RAW 264.7 (CL-0190) and THP-1 (CL-0233) cell lines were provided by the Institute of Stomatological Research, Sun Yat-sen University. The cells were treated with BBR solution at final concentrations of 200 µg/mL, 100 µg/mL, and 50 µg/mL under 5% CO₂ at 37°C for 0.5 h and 8 h to evaluate the short-term and long-term effects of BBR. After incubation, CCK-8 reagent (Cell-Counting-Kit 8, Gbcbio, China) was added to each well. The plate was then incubated at 37°C for 2 h, and the optical density at 450 nm (OD₄₅₀) of each well was measured using a microplate reader (Epoch2, BioTek, USA). The CCK-8 assay was performed in triplicate.

What's more, we have added 5-minute and 10-minute treatment data to simulate clinical applications like mouthwashes or toothbrushing. The results of CFU counting assay showed that 100 µg/mL and 200 µg/mL of BBR had comparable short-term antibacterial effect.

We fully acknowledge that our study has limitation on the minimum treatment time required to achieve an antimicrobial effect against *S. mutans*. We will absolutely implement more experiments in future studies to thoroughly evaluate the effect of BBR. We hope this supplemental data adequately addresses your concern. Thank you for this constructive suggestion that will undoubtedly improve our future work.

The other revised section now reads:

2.1. BBR inhibited *S. mutans* planktonic growth and biofilm formation

The MIC and MBC of BBR against planktonic *S. mutans* were 100 µg/mL assessed by in vitro test (Figure 1A). The growth curve assay, which could show the proliferation of bacteria directly, demonstrated a dose-dependent inhibitory effect of BBR on the proliferation of *S. mutans* planktonic cells, with 100 µg/mL and 200 µg/mL of BBR completely inhibiting bacterial proliferation (Figure 1B, 1C, 1D). BBR was found to inhibit the growth of *S. gordonii* DL-1 and *S. mitis* ATCC 6249 to some extent (Figure 1E, 1F). Moreover, we treated *S. mutans* UA159 with BBR solutions for 5 and 10 minutes. The results of CFU counting assay showed that 100 µg/mL and 200 µg/mL of BBR had comparable short-term antibacterial effect.

Figure 1. Inhibitory effect of BBR on the growth of bacteria. (A) The MIC and MBC values of BBR against planktonic bacteria. 106 CFU/mL bacteria in BHI broth were incubated with final concentrations of BBR ranging from 25-400 $\mu\text{g/mL}$ anaerobically at 37°C for 24 h. The MIC was defined as the lowest concentration of BBR that inhibited visible bacterial growth. At the termination of the MIC assay, a volume of the culture was struck on BHIA and incubated to observe growth. The MBC was defined as the lowest concentration that yielded no colony growth by subculturing on BHIA plates. (B-F) The 24 h growth curve of planktonic *S. mutans* UA159, *S. mutans* ATCC 25175, *S. mutans* GS-5, *S. gordonii* DL-1 and *S. mitis* ATCC 6249 incubated under anaerobic conditions with/without treatment of BBR in a 96-well plate. The absorbance of each well was recorded every hour. (G-H) The short-term antibacterial effect of BBR on *S. mutans* UA159 was assessed by measuring the average number of CFU. *S. mutans* UA159 was treated with BBR for 5 minutes (G) and 10 minutes (H). The suspensions were then diluted and plated onto BHIA plates and incubated under anaerobic conditions at 37°C for 24 h to determine CFU counts. Values represent the means \pm SD from three independent experiments (** $p < 0.01$, *** $p < 0.001$).

3. Discussion

The study has several limitations. Firstly, the study focuses on the in-vitro effects of BBR against *S. mutans*, which may not accurately reflect the in-vivo effects of BBR in clinical application due to the complex inter-action of oral microbiota and the influence of oral environment. While *S. mutans* seem to be sensitive to BBR, future studies should further investigate the effects of BBR against other bacteria and multi-strain biofilms to clarify the mechanism of BBR more comprehensively. Secondly, considering the favorable antimicrobial effect of BBR on *S. mutans*, BBR is expected to play an anticaries role as a kind of gargle or toothpaste additive, which

requires further research to investigate the minimum effective time and long-term biosafety in vivo. In addition, an effective concentration of BBR, in the presence of resveratrol, could be decreased even to 50% in cancer treatment (56), which suggests that the application of BBR in synergy with other agents may enhance the effects of BBR. Therefore, strategies to enhance the antimicrobial efficacy of BBR, including the combination of BBR with other antimicrobial agents or natural compounds to enhance its anticaries effects, would contribute to the clinical application of BBR. Notwithstanding these limitations, the study has important implications for the potential clinical application of BBR in the prevention of dental caries and may contribute to the development of novel anticaries approaches.

4.10. CFU counting assay

The CFU counting assay was used to quantify the impact of BBR on biofilm formation and mature bio-film of *S. mutans* by counting the number of live bacteria in *S. mutans* biofilm as reported (62). For the biofilm formation assay, *S. mutans* was incubated following the procedure outlined in section 4.4. After incubation, the adherent *S. mutans* cells in biofilms were resuspended and serially diluted from 10⁴-fold to 10⁶-fold. The suspensions were then plated onto BHIA plates and incubated under anaerobic conditions at 37°C for 48 h to determine CFU counts. When it came to the mature biofilm assay, overnight cultures of *S. mutans* were diluted in BHIS and incubated under anaerobic conditions at 37°C for 24 h to form mature biofilms. After that, BBR solution was added to each well to achieve final concentrations of 200 µg/mL, 100 µg/mL, and 50 µg/mL. The plate was then incubated under anaerobic conditions at 37°C for another 24 h followed by the same procedures as described above to determine CFU counts in each well. In order to evaluate the short-term effect of BBR, *S. mutans* UA159 was diluted to 10⁶ CFU/mL and treated with BBR solutions of varying final concentrations (200 µg/mL, 100 µg/mL, and 50 µg/mL) for 5 and 10 minutes. The suspensions were then diluted and plated onto BHIA plates and incubated under anaerobic conditions at 37°C for 24 h to determine CFU counts. The CFU counting procedures were performed in triplicate.

12. Finally, the authors are strongly encouraged to present biocompatibility data using human macrophages, such as the THP-1 cell line, instead of the murine macrophage cell line currently employed, to model the human immune response better.

Response:

We sincerely appreciate your thoughtful suggestions. We have added THP-1 cell lines to our experiments in order to address the concerns. These modifications significantly strengthen the clinical relevance of our biocompatibility assessment. We thank you for these valuable suggestions that have enhanced our study.

The revised section now reads:

2.6. BBR exhibited good biocompatibility

In order to evaluate the biocompatibility of BBR in the oral cavity, we treated human gingival epithelial cells (HGECs), human oral keratinocytes (HOKs), human periodontal ligament cells

(HPDLCs), human gingival fibroblasts (HGFs), RAW 264.7 and THP-1 with varying concentrations of BBR for 0.5 h and 8 h to evaluate its short-term and long-term effects, and measured cell viability using a CCK-8 assay (Figure 8). HGECs and HOKs represented the oral mucosa which is in direct contact with the BBR, while HPDLCs, HGFs, RAW 264.7 and THP-1 which is not in direct contact with BBR. When treated with BBR for 0.5 h, none of the tested concentrations of BBR exhibited a significant inhibitory effect on the cell viability of HOKs, HGFs, RAW 264.7 or THP-1. However, the concentration of 200 $\mu\text{g}/\text{mL}$ of BBR slightly decreased the cell viability of HGECs and HPDLCs. Moreover, 200 $\mu\text{g}/\text{mL}$ of BBR inhibited the cell viability of HGECs, HOKs, HPDLCs, HGFs, RAW 264.7 and THP-1 when treated for 8 h, while 50 $\mu\text{g}/\text{mL}$ and 100 $\mu\text{g}/\text{mL}$ of BBR exhibited not statistically significant effect. This indicates that BBR has good biocompatibility when applied to cultured cells to a certain extent.

Figure 8. Biocompatibility of BBR evaluated by CCK-8 assay. The assay measured the effect of BBR on the cell viability of HGECs, HOKs, HPDLCs, HGFs, RAW 264.7 and THP-1. The values represent the means \pm SD from three independent experiments (* $p < 0.05$, ns: not statistically significant compared to the untreated control group).

4.15. Cytotoxicity effect on host cells

The biocompatibility of BBR on human gingival epithelial cells (HGECs), human oral keratinocytes (HOKs), human periodontal ligament cells (HPDLCs), human gingival fibroblasts (HGFs), and macrophages (RAW 264.7 and THP-1) was evaluated by the Cell-Counting-Kit 8 (CCK-8) assay following the manufacturer's instructions. The HGEC (CP-H178), HOK (CP-H382), HPDLC (CP-H234), HGF (CP-H240), RAW 264.7 (CL-0190) and THP-1 (CL-0233) cell lines were provided by the Institute of Stomatological Research, Sun Yat-sen University. The cells were treated with BBR solution at final concentrations of 200 µg/mL, 100 µg/mL, and 50 µg/mL under 5% CO₂ at 37°C for 0.5 h and 8 h to evaluate the short-term and long-term effects of BBR. After incubation, CCK-8 reagent (Cell-Counting-Kit 8, Gbcbio, China) was added to each well. The plate was then incubated at 37°C for 2 h, and the optical density at 450 nm (OD₄₅₀) of each well was measured using a microplate reader (Epoch2, BioTek, USA). The CCK-8 assay was performed in triplicate.

Material and methods

13. "In Section 4.2, several points require clarification. The planktonic experiments are conducted using cultures in 96-well round-bottom cell culture plates. Should planktonic cultures instead be performed in tubes? In this plate type, *S. mutans* forms aggregates and deposits on the round bottom. Additionally, due to *S. mutans*'s ability to form biofilms, biofilm experiments for MIC and MBC determination must be conducted in 48-well or 24-well plates.

Response:

We appreciate your thoughtful comments regarding our experimental design. We would like to clarify several key points. In Section 4.2, we employed the standard two-fold serial dilution method for MIC/MBC assessment, with 96-well round-bottom cell culture plates commonly adopted in preliminary antimicrobial studies of natural products^[1-3]. Under this circumstance, the biofilm formation is prevented because that BHI lacks sufficient sucrose for biofilm formation within 24 hour and U-bottom geometry physically prevents surface attachment. We acknowledge that tube-based methods are an alternative approach, and the microplate method we used in this study are consistent with published antimicrobial studies as we mention before.

Additionally, the biofilm experiments are conducted in 24-well plates supplemented with 1% sucrose. We feel so sorry for our unclear explanation. We have added these clarifications to Section 4.4 to better explain our methodological choices. Thank you for prompting us to improve the manuscript.

[1] Ren, S., Yang, Y., Xia, M., Deng, Y., Zuo, Y., Lei, L., & Hu, T. (2023). A Chinese herb preparation, honokiol, inhibits *Streptococcus mutans* biofilm formation. *Archives of oral biology*, *147*, 105610. <https://doi.org/10.1016/j.archoralbio.2022.105610>

[2] Niu, Y., Wang, K., Zheng, S., Wang, Y., Ren, Q., Li, H., Ding, L., Li, W., & Zhang, L. (2020).

Antibacterial Effect of Caffeic Acid Phenethyl Ester on Cariogenic Bacteria and Streptococcus mutans Biofilms. *Antimicrobial agents and chemotherapy*, 64(9), e00251-20. <https://doi.org/10.1128/AAC.00251-20>

[3] Yue, J., Yang, H., Liu, S., Song, F., Guo, J., & Huang, C. (2018). Influence of naringenin on the biofilm formation of Streptococcus mutans. *Journal of dentistry*, 76, 24–31. <https://doi.org/10.1016/j.jdent.2018.04.013>

The revised section now reads:

4.4. Crystal violet staining assay

In order to evaluate the impact of BBR on *S. mutans* UA159 biofilm formation, crystal violet staining assay was performed following reported protocols (59-61) with modifications. Briefly, *S. mutans* UA159 was diluted to 10⁶ CFU/mL in BHIS and treated with BBR solutions of varying final concentrations (200 µg/mL, 100 µg/mL, and 50 µg/mL) in a 24-well flat-bottom plate under anaerobic conditions at 37°C for 24 h. After incubation, the biofilms were fixed using anhydrous methanol for 15 min. The fixed biofilms were stained with 0.1% (w/v) crystal violet for 5 min. After removing the solution, an anhydrous ethanol solution was added to each well to dissolve the dye under gentle shaking for 30 min. Finally, the optical density at 575 nm (OD₅₇₅) of each well was measured by a microplate reader (Epoch2, BioTek, USA). The assay was performed in triplicate.

14. Furthermore, as noted in the comments on the Results section, the exact MIC concentration should be precisely determined within the range of 50 to 100 µg/mL. The authors may consider testing at 5 µg/mL intervals (e.g., 55, 60, 65, etc.) until reaching 100 µg/mL."

Response:

We sincerely appreciate your insightful suggestion. In this study, we employed the standard two-fold serial dilution method for MIC/MBC assessment, with concentration intervals (50-100-200 µg/mL) commonly adopted in preliminary antimicrobial studies of natural products ^[1,2]. We acknowledge that finer concentration gradients (5-10 µg/mL intervals) would indeed provide more precise MIC values, as rightly suggested.

Moreover, in response to Reviewer #1's request, we have already expanded our experiments to include additional bacterial strains (*S. gordonii* and *S. mitis*) in the MIC/MBC assays. Implementing more refined concentration gradients across all tested strains would require repeating the entire experimental series, which unfortunately exceeds our current resource capacity in terms of both budget and personnel.

To demonstrate our commitment to addressing your valuable suggestion, we have conducted additional refined MIC/MBC testing specifically for *S. mutans* UA159 as a representative case in the response to you. We will absolutely implement finer concentration intervals in future studies to obtain more precise antimicrobial parameters.

We hope this supplemental data adequately addresses your concern. Thank you for this constructive suggestion that will undoubtedly improve our future work.

The additional data for *S. mutans* UA159 shows:

	MIC ($\mu\text{g/mL}$)	MBC ($\mu\text{g/mL}$)
S. mutans UA159	75	90

[1] Li J, Wu T, Peng W, Zhu Y. Effects of resveratrol on cariogenic virulence properties of *Streptococcus mutans*. *BMC Microbiol.* 2020;20(1):99. Published 2020 Apr 17. doi:10.1186/s12866-020-01761-3

[2] He Z, Huang Z, Jiang W, Zhou W. Antimicrobial Activity of Cinnamaldehyde on *Streptococcus mutans* Biofilms. *Front Microbiol.* 2019;10:2241. Published 2019 Sep 25. doi:10.3389/fmicb.2019.02241

15. The type of plates used for the crystal violet assay should be specified. Additionally, the authors appear to have obtained a stable *S. mutans* biofilm without adding sucrose. According to the literature, adding a minimal amount of sucrose (0.1%) is necessary for more stable biofilm formation. Did the authors assess the effects of the compound on more stable and mature biofilms, specifically those formed in the presence of sucrose?

It is well established that *S. mutans* utilizes sucrose to produce extracellular polysaccharides (EPS) via enzymes such as glucosyltransferases (Gtfs). These EPS form a sticky matrix that facilitates stronger adhesion to tooth surfaces and enhances bacterial cohesion. This matrix increases biofilm mass and promotes acid production, contributing to tooth decay, while offering protection against antimicrobial agents and host defenses.

Numerous studies have shown that *S. mutans* forms more robust and adhesive biofilms in the presence of sucrose than in the absence of other carbohydrates. Thus, sucrose is crucial in enhancing biofilm formation, adhesion, and virulence, increasing the risk of dental caries.

Response:

We appreciate your thoughtful comments regarding our experimental design. We would like to clarify several key points. In Section 4.4, the biofilm experiments are conducted in 24-well flat-bottom plates which is shown in Figure 2A. Additionally, the biofilm experiments are conducted supplemented with 1% sucrose which is shown in Section 4.1 and Section 4.4 that:

4.1. Chemicals, bacterial strain, and growth conditions

BHI supplemented with 1% (w/v) of sucrose (BHIS) was used as culture medium in biofilm assays.

4.4. Crystal violet staining assay

Briefly, *S. mutans* UA159 was diluted to 10^6 CFU/mL in BHIS and treated with BBR solutions of varying final concentrations (200 $\mu\text{g/mL}$, 100 $\mu\text{g/mL}$, and 50 $\mu\text{g/mL}$) in a 24-well flat-bottom plate under anaerobic conditions at 37°C for 24 h.

We feel so sorry for our unclear description. We have added these clarifications to better explain

our methodological choices. Thank you for prompting us to improve the manuscript.

The revised section now reads:

4.1. Chemicals, bacterial strain, and growth conditions

Berberine hydrochloride (hereafter referred to as berberine, BBR) used in this study was obtained from Macklin Inc. (B796571, $\geq 99\%$). BBR was dissolved in double distilled water and filtered before used. The bacterial strain *S. mutans* UA159 was obtained from the American Type Culture Collection (ATCC). The bacteria were recovered overnight at 37°C and diluted in Brain Heart Infusion (BHI) broth (HuanKai Microbial, China) for further experiments. BHI was used as culture medium in planktonic bacteria assays. BHI supplemented with 1% (w/v) of sucrose (BHIS) was used as culture medium in biofilm assays. BHI supplemented with 1% (w/v) of agar (BHIA) was used in CFU counting assays. The negative control group was set as double distilled water of the same volume as BBR.

4.4. Crystal violet staining assay

In order to evaluate the impact of BBR on *S. mutans* UA159 biofilm formation, crystal violet staining assay was performed following reported protocols (59-61) with modifications. Briefly, *S. mutans* UA159 was diluted to 106 CFU/mL in BHIS and treated with BBR solutions of varying final concentrations (200 $\mu\text{g/mL}$, 100 $\mu\text{g/mL}$, and 50 $\mu\text{g/mL}$) in a 24-well flat-bottom plate under anaerobic conditions at 37°C for 24 h. After incubation, the biofilms were fixed using anhydrous methanol for 15 min. The fixed biofilms were stained with 0.1% (w/v) crystal violet for 5 min. After removing the solution, an anhydrous ethanol solution was added to each well to dissolve the dye under gentle shaking for 30 min. Finally, the optical density at 575 nm (OD₅₇₅) of each well was measured by a microplate reader (Epoch2, BioTek, USA). The assay was performed in triplicate.

16. Did the authors compare the pH drop at higher versus lower glucose concentrations?

Response:

We sincerely appreciate your insightful comment. In this study, we selected the commonly used glucose concentration of 1% (w/v) for pH drop assays, as established in prior literature ^[1,2]. While the distinction between "high" and "low" glucose concentrations may vary across studies without a unified standard, we fully acknowledge the value of this comparative perspective. To address your concern and further strengthen our work, we have now supplemented additional experiments using 0.1% glucose as a lower concentration. The results demonstrate that the pH reduction at 0.1% glucose is indeed less pronounced overall, particularly showing a significantly attenuated initial drop compared to the 1% glucose condition. We hope these additional data provide further clarity regarding concentration-dependent effects. Thank you for this constructive suggestion.

[1] He, Z., Huang, Z., Jiang, W., & Zhou, W. (2019). Antimicrobial Activity of Cinnamaldehyde on *Streptococcus mutans* Biofilms. *Frontiers in microbiology*, 10, 2241. <https://doi.org/10.3389/fmicb.2019.02241>

[2] Mu, R., Zhang, H., Zhang, Z., Li, X., Ji, J., Wang, X., Gu, Y., & Qin, X. (2023). Trans-cinnamaldehyde loaded chitosan based nanocapsules display antibacterial and antibiofilm effects against cavity-causing *Streptococcus mutans*. *Journal of oral microbiology*, *15*(1), 2243067. <https://doi.org/10.1080/20002297.2023.2243067>

The revised section now reads:

2.3. BBR inhibited *S. mutans* virulence factors

The ability to synthesize EPS is one of the main virulence factors *S. mutans*. EPS aid in the permanent colonization of hard surfaces and in the development of the extracellular polymeric matrix in situ (24). CLSM, which could observe the three-dimensional images of the *S. mutans* biofilm, was used to quantify the amounts of bacteria and EPS synthesized by *S. mutans* under the treatment of BBR. The biofilms were cultured on glass coverslips for efficient image capture and data collection. BBR was added immediately after bacterial dilutions were added to the culture plate, and the images were captured 24 h later. Figure 5A shows a series of three-dimensional micro-images of *S. mutans* biofilms formed on glass coverslips, captured by CLSM. The bacteria and EPS were labeled green and red, respectively. Image J COMSTAT was used to analyze the distribution of bacteria and EPS in each layer of the *S. mutans* biofilms, as well as the relevant biomass. Moreover, the total biomass of bacteria and EPS in *S. mutans* biofilms was calculated according to the statistics. In the absence of BBR, the biofilm exhibited a uniform distribution with a relatively dense structure and complete surface coverage (Figure 5B). Following treatment with BBR, the surface area covered by the biofilm decreased visibly, resulting in a significant reduction in biofilm biomass (Figure 5C). The micro-images and statistical results demonstrate that bacterial proliferation and EPS synthesis were inhibited in the BBR-treated groups, resulting in a decrease in the accumulation of *S. mutans* adherent cells and EPS. Moreover, treatment with varying concentrations of BBR led to a significant reduction in the synthesis of water-insoluble EPS (Figure 5D) in *S. mutans* biofilm, which is consistent with the result of SEM. Acidogenicity is another main virulence factor of *S. mutans*. The impact of BBR on acid production by *S. mutans* was evaluated using the lactic acid assay and glycolytic pH drop assay. *S. mutans* was cultured in the presence of varying concentrations of BBR, and the production of lactic acid was measured. Figure 5E showed that BBR decreased the lactic acid production of the biofilms compared to the control group. However, acid production does not inevitably lead to dental erosion. The critical pH of dental enamel is approximately 5.5 and any solution with a lower pH value may cause erosion (25). Therefore, the glycolytic pH drop assay was further conducted. As shown in Figure 5F and Figure 5G, the pH values of the BBR treated group significantly differed from those of the control group. In general, BBR reduced the rate of pH drop compared to the control group. In 1% glucose-supplemented group, the final pH values decreased from 7.22 ± 0.02 to 4.98 ± 0.09 in the control group after 270 min of incubation. Treatment with BBR at concentrations of 50 $\mu\text{g/mL}$, 100 $\mu\text{g/mL}$ and 200 $\mu\text{g/mL}$ increased the terminal pH to 5.97 ± 0.02 , 6.74 ± 0.01 , and 6.86 ± 0.01 , respectively. The results showed that the control group experienced the greatest decrease in pH within the first 30 min of incubation, dropping from 7.22 ± 0.02 to 6.34 ± 0.05 . In contrast, the group treated with 200 $\mu\text{g/mL}$ BBR experienced the smallest drop in pH, from 6.86 ± 0.01 to 7.11 ± 0.02 (Figure 5F). The pH reduction in 0.1% glucose-supplemented group is less pronounced overall, particularly showing a significantly attenuated initial drop compared to the 1%

glucose-supplemented group (Figure 5G). The results indicated that BBR inhibits *S. mutans* acidogenicity and alleviates pH drop.

Figure 5. BBR inhibited *S. mutans* virulence factors. (A) Effect of BBR on the biofilm structure of *S. mutans* observed by CLSM. Double-labeling imaging of *S. mutans* 24 h biofilm formed on glass coverslips. The fluorescence (SYTO 9) marks the live bacteria, while the red fluorescence (Concanavalin A-TRITC) marks the EPS synthesized by *S. mutans*. (B) Quantification of the amounts of EPS and bacteria in each scanned layer of *S. mutans* 24 h biofilm without the treatment of BBR. (C) Quantification of the amounts of EPS and bacteria in each scanned layer of *S. mutans* 24 h biofilm with the treatment of 50 µg/mL BBR. (D) Quantitative measurement of water-insoluble EPS by anthrone-sulfuric method. (E) Measurement of lactic acid production. (F-G) Effect of BBR on *S. mutans* glycolytic pH drop under 1% (F) and 0.1% (G) glucose. Values represent the means \pm SD from three independent experiments (*** $p < 0.001$).

4.7. Glycolytic pH drop assay

The impact of BBR on the pH drop of *S. mutans* glycolysis was measured by glycolytic pH drop assay. Briefly, *S. mutans* was harvested at mid-logarithmic phase via centrifugation, washed with a 50 mM KCl solution, and resuspended in the same salt solution. Glucose was added to each tube to achieve a final concentration of 0.1% and 1% (w/v). BBR was then added to each tube to achieve final concentrations of 200 µg/mL, 100 µg/mL, and 50 µg/mL. A suspension of *S. mutans* in salt solution was used as a negative control. The pH decrease resulting from the glycolytic activity of *S. mutans* was monitored at 30-minute intervals over a period of 270 min using a pH meter (FE28, METTLER TOLEDO, Switzerland). The assay was performed in triplicate.

17. Throughout the manuscript, the live staining is inconsistently referred to as either SYTO9 or SYTO 9. It should be standardized to 'SYTO 9' only.

Response:

We sincerely appreciate your careful attention to detail. As suggested, we have standardized the terminology to "SYTO 9" throughout the manuscript to ensure consistency. Thank you for this helpful correction, which improves the precision of our reporting.

18. Section 2.15 should be renamed 'The BBR Cytotoxicity Effect on Host Cells.' Additionally, as mentioned earlier, why were the cells only treated for 30 minutes? To assess long-term effects, the authors should consider incorporating longer treatment durations. Additional experiments with extended treatment times would be valuable to explore the potential future use of this compound in the oral cavity. Furthermore, the choice of assay should be discussed in more detail.

Response:

We sincerely appreciate your insightful suggestions regarding treatment duration. In response to these valuable comments, we have renamed the title of the section and conducted additional experiments to more comprehensively evaluate the effects of BBR. We have added 8-hour treatment data to simulate overnight exposure conditions. The results showed that treatment with BBR for 0.5 h did not significantly affect the cell viability of HOKs, HGFs, RAW 264.7 or THP-1, while a slight inhibition of cell viability was observed in HGECs and HPDLCs at a concentration of 200 µg/mL of BBR. Moreover, 200 µg/mL of BBR inhibited the cell viability of HGECs, HOKs, HPDLCs, HGFs, RAW 264.7 and THP-1 when treated for 8 h, while 50 µg/mL and 100 µg/mL of BBR exhibited not statistically significant effect, indicating that long exposure to high concentration of BBR would affect the cell viability. The antibacterial effects and cytotoxicity of BBR show consistency. While 200 µg/mL of BBR serves as a higher concentration with enhanced antibacterial effects, it also increases cytotoxicity, highlighting the importance of concentration selection for antibacterial agents. Given that 100 µg/mL of BBR exhibited comparable short-term antibacterial effect (Figure 1G, 1H) and favorable anti-caries effects with not statistically significant effect on the viability of HGECs, HOKs, HPDLCs, HGFs, RAW 264.7 and THP-1, BBR is a biocompatible and suitable option for clinical use as a novel anti-caries agent.

We believe these additions significantly strengthen our findings while addressing your concerns. Thank you for this suggestion that has enhanced our study.

The revised section now reads:

2.6. BBR exhibited good biocompatibility

In order to evaluate the biocompatibility of BBR in the oral cavity, we treated human gingival epithelial cells (HGECs), human oral keratinocytes (HOKs), human periodontal ligament cells (HPDLCs), human gingival fibroblasts (HGFs), RAW 264.7 and THP-1 with varying concentrations of BBR for 0.5 h and 8 h to evaluate its short-term and long-term effects, and measured cell viability using a CCK-8 assay (Figure 8). HGECs and HOKs represented the oral mucosa which is in direct contact with the BBR, while HPDLCs, HGFs, RAW 264.7 and THP-1 which is not in direct contact with BBR. When treated with BBR for 0.5 h, none of the tested concentrations of BBR exhibited a significant inhibitory effect on the cell viability of HOKs, HGFs, RAW 264.7 or THP-1. However, the concentration of 200 $\mu\text{g}/\text{mL}$ of BBR slightly decreased the cell viability of HGECs and HPDLCs. Moreover, 200 $\mu\text{g}/\text{mL}$ of BBR inhibited the cell viability of HGECs, HOKs, HPDLCs, HGFs, RAW 264.7 and THP-1 when treated for 8 h, while 50 $\mu\text{g}/\text{mL}$ and 100 $\mu\text{g}/\text{mL}$ of BBR exhibited not statistically significant effect. This indicates that BBR has good biocompatibility when applied to cultured cells to a certain extent.

Treated with BBR for 0.5 h

Treated with BBR for 8 h

Figure 8. Biocompatibility of BBR evaluated by CCK-8 assay. The assay measured the effect of BBR on the cell viability of HGECs, HOKs, HPDLCs, HGFs, RAW 264.7 and THP-1. The values represent the means \pm SD from three independent experiments (* $p < 0.05$, ns: not statistically significant compared to the untreated control group).

3. Discussion

When evaluating a new agent for clinical use, it is important to consider its biocompatibility. In this study, the biocompatibility of BBR was evaluated by measuring its effect on the cell viability of HGECs, HOKs, HPDLCs, HGFs, RAW 264.7 and THP-1. HGECs and HOKs represented the oral mucosa in direct contact with BBR, while HPDLCs, HGFs, RAW 264.7 and THP-1 represented the submucosal tissues. The results (Figure 8) showed that treatment with BBR for 0.5 h did not significantly affect the cell viability of HOKs, HGFs, RAW 264.7 or THP-1, while a slight inhibition of cell viability was observed in HGECs and HPDLCs at a concentration of 200 $\mu\text{g/mL}$ of BBR. Moreover, 200 $\mu\text{g/mL}$ of BBR inhibited the cell viability of HGECs, HOKs, HPDLCs, HGFs, RAW 264.7 and THP-1 when treated for 8 h, while 50 $\mu\text{g/mL}$ and 100 $\mu\text{g/mL}$ of

BBR exhibited not statistically significant effect, indicating that long exposure to high concentration of BBR would affect the cell viability. The antibacterial effects and cytotoxicity of BBR show consistency. While 200 µg/mL of BBR serves as a higher concentration with enhanced antibacterial effects, it also increases cytotoxicity, highlighting the importance of concentration selection for antibacterial agents. Given that 100 µg/mL of BBR exhibited comparable short-term antibacterial effect (Figure 1G, 1H) and favorable anti-caries effects with not statistically significant effect on the viability of HGECs, HOKs, HPDLCs, HGFs, RAW 264.7 and THP-1, BBR is a biocompatible and suitable option for clinical use as a novel anticaries agent.

4.15. Cytotoxicity effect on host cells

The biocompatibility of BBR on human gingival epithelial cells (HGECs), human oral keratinocytes (HOKs), human periodontal ligament cells (HPDLCs), human gingival fibroblasts (HGFs), and macrophages (RAW 264.7 and THP-1) was evaluated by the Cell-Counting-Kit 8 (CCK-8) assay following the manufacturer's instructions. The HGEC (CP-H178), HOK (CP-H382), HPDLC (CP-H234), HGF (CP-H240), RAW 264.7 (CL-0190) and THP-1 (CL-0233) cell lines were provided by the Institute of Stomatological Research, Sun Yat-sen University. The cells were treated with BBR solution at final concentrations of 200 µg/mL, 100 µg/mL, and 50 µg/mL under 5% CO₂ at 37°C for 0.5 h and 8 h to evaluate the short-term and long-term effects of BBR. After incubation, CCK-8 reagent (Cell-Counting-Kit 8, Gbcbio, China) was added to each well. The plate was then incubated at 37°C for 2 h, and the optical density at 450 nm (OD₄₅₀) of each well was measured using a microplate reader (Epoch2, BioTek, USA). The CCK-8 assay was performed in triplicate.

Re: Spectrum00700-25R1 (Inhibitory effects of berberine against *Streptococcus mutans*: an in vitro insight on its anticaries potential)

Dear Prof. Yang Cao:

Your manuscript has been accepted, and I am forwarding it to the ASM production staff for publication. Your paper will first be checked to make sure all elements meet the technical requirements. ASM staff will contact you if anything needs to be revised before copyediting and production can begin. Otherwise, you will be notified when your proofs are ready to be viewed.

Sincerely,
Sébastien Faucher
Editor
Microbiology Spectrum